

# Response of tidal flow regime and sediment transport in North Male' Atoll, Maldives to coastal modification and sea level rise

Shuaib Rasheed [1], Simon C. Warder [1], Yves Plancherel [1,2], and Matthew D. Piggott [1]

[1]Department of Earth Science and Engineering, Imperial College London, UK
[2]Grantham Institute – Climate Change and the Environment, Imperial College London, UK

**Correspondence:** Shuaib Rasheed (s.rasheed18@imperial.ic.ac.uk)

**Abstract.** Changes to coastlines and bathymetry alter tidal dynamics and associated sediment transport process, impacting upon a number of threats facing coastal regions, including flood risk and erosion. Especially vulnerable are coral atolls such as those that make up the Maldives archipelago which has undergone significant land reclamation in recent years and decades, and is also particularly exposed to sea level rise. Here we develop a tidal model of Male' Atoll, Maldives, and use it to assess potential changes to sediment grain size distributions under sea level rise and coastline alteration scenarios. The results indicate that the impact of coastline modification over the last two decades at the island scale is not limited to the immediate vicinity of the modified island, but can also significantly impact the sediment grain size distribution across the wider atoll basin. Additionally, the degree of change in sediment distribution which can be associated with sea level rise that is projected to occur over relatively long time periods is predicted to occur over far shorter time periods with coastline changes, highlighting the need to better understand, predict and mitigate the impact of land reclamation and other coastal modifications before conducting such activities.

## 1 Introduction

Driven by their importance to the coastal zone, the response of tidal dynamics and sediment to future sea level rise scenarios as well as to coastal modification has been considered in various studies targeting different locations around the world, ranging from marginal seas such as the Bohai Sea (e.g., Pelling et al., 2013), shelf seas such as the North-West European continental shelf (e.g., Ward et al., 2012) and estuaries and bays such as the Eastern Scheldt estuary (e.g., Jiang et al., 2020). However, the response of sediment distribution in large coral atolls to anthropogenic pressures such as land reclamation and coastal modification remains poorly studied and is generally restricted to very small patch reefs for a variety of reasons including remoteness and lack of data. Recent availability of a high-resolution bathymetry dataset for the Maldives archipelago (Rasheed et al.) now allows for the application of numerical models capable of studying the hydrodynamics within the large coral atolls of the archipelago at high fidelity for the first time.

The relationship between tidal currents and sediment distribution patterns has been considered as early as Kenyon (1970) through field observations. Pingree and Griffiths (1979) used a numerical model to derive a correlation between sand transportation pathways and bed shear stress derived from the combined M2 and M4 tidal constituents in the shelf seas around the





UK. Warwick and Uncles (1980) used tidal bed shear stress derived from a numerical model to infer benthic sediment in the Bristol Channel and identified the issue of overlapping bed shear stress values derived from numerical models and multiple bed sediment types. Ward et al. (2015) further developed a classification scheme for bed shear stress derived from a numerical model and correlated it with the observed dominant seabed sediment type, resolving the issue of overlapping bed sediment types and modelled bed shear stress values.

30       Studies (e.g., Holleman and Stacey, 2014; Jiang et al., 2020) have shown that the response of the tides to changing physical characteristics such as bathymetry and coastline modification can vary significantly in different areas depending on the local geological setting. According to these studies the response in some environments is more complex than others due to the effects of processes such as shoaling, damping and resonance. Holleman and Stacey (2014) reports significant tidal amplification increases resulting from sea level rise in San Pablo bay, California due to the complex bathymetry of the region. Similarly,

coral atolls, described as being analogous to large "leaky buckets" Gischler (2006) composed of drowned carbonate platforms with extremely complex topographic features such as steep vertically rising lagoons, channels and oceanic faros, will be subject to similarly complex responses. This is supported by field data from different sources including Betzler et al. (2015); Gischler (2006); Morgan and Kench (2014), who studied the bed sediment in different areas of the Maldives archipelago at different scales ranging from regional (e.g. atoll) scales to localised (e.g. individual island/lagoon) scales.

40       The aim of this study is to construct a classified bed sediment map of North Male' atoll, Maldives, using tidal simulations validated against available tidal and sediment field measurements, and to use this tool to quantify the response to sea level rise and large-scale land reclamation scenarios. The strong correlation reported between velocity patterns and dominant bed sediment grain size in coral atolls provide grounds and data to derive an estimated grain size distribution from the outputs of tidal simulations, which can be further used to classify the bed shear stress at the atoll level and also to understand the potential

response of atoll systems to anthropogenic pressures. Identification of potential bed sediment type is important for a variety of reasons ranging from the identification of potential dredging sites, the determination of water turbidity and the identification of potential habitats for benthic flora and fauna. The methods described in this study can be applied to other coral atolls of the Maldives and other regions elsewhere with similar geological settings and where there are limitations in accessing field data.

## 2   Study Site

### 2.1   General Setting

The Maldives archipelago, shown in Figure 1 (a), is located to the South-West of the Indian subcontinent. Bounded by the 2000 m bathymetric contour of the Chagos-Laccadive ridge, the archipelago ranges over approximately 1000 km from north to south and 150 km from east to west. The 22 atolls that form the Maldives archipelago each have their own unique characteristics and range in size from a few kilometres to tens of kilometres (Wells, 1988), encompassing thousands of individual reefs and more

than 1200 low-lying islands (Naseer and Hatcher, 2004). None of the islands exceed more than a few metres in height above current sea level. The most prominent feature of the Maldives archipelago is the arrangement of the double chain of atolls in the central zone of the archipelago separated by the Maldives inner sea with depths typically in the range 300–500 m (Purdy and



Bertram, 1993). Even though there have been a limited number of studies, the geological features of Maldives coral atolls and their formation have been discussed since observations obtained from field expeditions in the late 19[th] and early 20[th] centuries (e.g., Gardiner, 1902, 1903; Glennie, 1936; Hass, 1965) as well as more recent studies in the past few decades (e.g., Aubert and Droxler, 1992, 1996; Purdy and Bertram, 1993; Belopolsky and Droxler, 2003; Belopolsky et al., 2004)

Located in the doldrums, the Maldives does not generally experience major storms and the climate is primarily influenced by the seasonal fluctuations of the South Asian monsoonal wind patterns, with the wind speed averaging 5 m s$^{-1}$ for both the north-east and south-western monsoons. The archipelago experiences a semidiurnal microtidal regime with a tidal range of approximately 1 m (Caldwell et al., 2015). The combined tidal- and wind-driven currents can exceed speeds of 2 m s$^{-1}$, particularly in the channels separating the atolls, along the ocean-facing flanks of atoll rims and through gaps in atoll rims (Ciarapica and Passeri, 1993).

## 2.2   Male' Atoll

Male' atoll is an administrative unit in the Maldives archipelago located on the north-eastern side of the double chain of atolls of the Maldives archipelago, and is comprised of the geographic atolls of Kaashidoo, Gaafaru, and North and South Male' atolls. Kaashidhoo atoll, classified as an oceanic platform reef, is separated from the other parts of the administrative unit by deep channels exceeding 500 m separating the atoll on all sides. Due to the distance from the main geographical areas of the administrative atoll and its distinct nature, Kaashidhoo atoll was not included in this study.

Gaafaru atoll, seen in Figure 1 (b), also classified as an oceanic faro (Naseer and Hatcher, 2004), is a small atoll 15 km in width and 8 km in length with a surface area of 88.50 km$^2$. The atoll, similar to other atolls classified as oceanic faros across the country, is significantly smaller than the larger more complex atolls. The outer rim of Gaafaru atoll has no major openings besides two channels in the North of the atoll. The only island in the atoll is Gaafaru island to the south-east and no other lagoon or faro exists in the atoll. The atoll is separated from North Male' atoll by the narrow, deep channel *Hani Kandu* which is 3 km wide with depths of up to 160 m in the central region of the channel (Luthfee, 1995). We include Gaafaru atoll in the simulations conducted here because field studies (Gischler, 2006) suggest that smaller atolls of the archipelago have different characteristics compared to the larger more complex atolls in terms of their response to changes in tides and wind patterns.

North Male' atoll and South Male' atoll, seen in Figure 1 (b), are both large complex atolls making up the bulk of the area of the administrative unit of Male' atoll. The larger North Male' atoll measures 60 km in length and 40 km in width with a surface area of ~1623.92 km$^2$. The atoll contains more than 189 individual reefs for a total reef area of 349 km$^2$ (Naseer and Hatcher, 2004). The atoll has numerous faros, particularly concentrated in the south and the Northern regions of the atoll, with the faros found in the southern zone being more shallow and prominent. The atoll also consists of numerous islands, both natural and reclaimed. The southern islands of the atoll, which include the major islands of Male' (the capital island of the country), Hulhule', Hulhumale' and Vilingilli, are home to more than 150,000 people, which represents more than a third of the population of the entire country. To accommodate the socio-economic changes that have occurred over the past few decades, fuelled largely by the tourism industry, almost all islands in the atoll have experienced coastal modifications in terms of both harbour construction and land reclamation activities, with nearly half of all the currently existing land being reclaimed





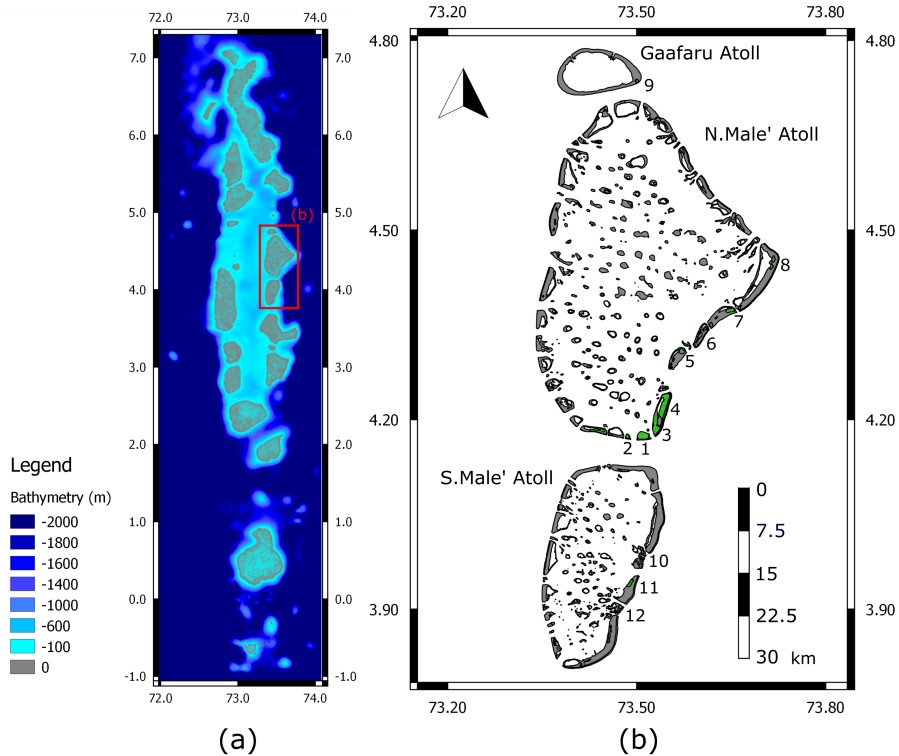

**Figure 1.** (a) Bathymetry of the Maldives archipelago as per Rasheed et al.. (b) The study region which includes the atolls of Gaafaru atoll, North Male' atoll and South Male' atoll with the lagoons (Grey) and islands (Green) marked. Inhabited islands, excluding industrial islands, are labelled as 1:Male', 2:Villingilli, 3:Hulhule', 4:Hulhumale', 5:Himmafushi, 6:Huraa, 7:Thulusdhoo, 8:Dhiffushi, 9:Gaafaru, 10:Gulhi, 11:Maafushi, 12:Guraidhoo); boundaries of lagoons based upon Spalding et al. (2001).

(Duvat and Magnan, 2019). This makes the location an ideal setting within which to consider the impact of such anthropogenic modifications.

95    South Male' atoll is separated from North Male' atoll by a 5 km wide channel *Vaadhoo Kandu*, with depths of close to 400 m in its central regions (Luthfee, 1995). South Male' atoll has a surface area of ∼558.31 km$^2$. The atoll also has numerous reefs with 112 individual reefs identified, (Naseer and Hatcher, 2004), with a combined area of ∼175.60 km$^2$. However, unlike North Male' atoll where the outer rim is cut with numerous channels, the outer rim of South Male' atoll is more continuous with fewer channels separating the outer rim. While South Male' atoll has also seen extensive land reclamation, these activities at large scale have occurred only in the last few years and the changes are thus not included in this study. However, sediment

100    grain size data available for the atoll (Betzler et al., 2016) was used as the main source of observational data to derive the grain size tidal proxy.



## 2.3 Sediment Data

The bed sediment of the coral atolls of the Maldives archipelago has been examined by several studies at the local scale across individual reefs, as well as at a regional scale spanning one or more atolls, mainly for the purposes of better understanding the formation of the archipelago (Ciarapica and Passeri, 1993). However, the correlation between bed sediment type and flow patterns in the Maldives archipelago has been known since Darwin (1842). With the sediment entirely devoid of terrigenous input due to the distance from major landmasses over the past 55 My (Belopolsky and Droxler, 2003; Aubert and Droxler, 1992), most bed sediment studies in the archipelago have focused on identifying the biological composition of the benthic surface, with a minimum number of studies focusing on sedimentary dynamics, especially at regional (atoll) scales (e.g., Betzler et al., 2015; Naseer, 2003).

### 2.3.1 Localized (Island/Lagoon scale Data)

The sediment patterns around Funadhoo island in North Male' atoll, described by Kohn (1964), indicate a sediment pattern dominated heavily by wind waves. Located on the channel between Male' and Hulhule', the island is exposed to continuous heavy swells, with thin layers of finer particles such as sand confined to areas of the lagoon that are less exposed. This observation of wind-driven sediment patterns in the shallow lagoons of the Maldives archipelago was further studied by Kench et al. (2009) and Kench and Brander (2006), who reported large movements of sand horizontally around islands due to changes in monsoonal wind patterns. Further, quantitative studies of sediment transport patterns in Vabbinfaru reef in North Male' atoll by Morgan and Kench (2014) between two monsoonal periods showed that significant quantities of sediment are transported from the lagoon to the atoll basin, with the main mode of transportation identified as wind-driven waves associated with the South West monsoon. These studies suggest a wind dominated sediment transport pattern for shallow lagoon areas of the Maldives archipelago. However, as discussed in the next section, observational data from the atoll basin which makes up the bulk of the atoll area, including marine benthic fauna and sediment data, indicate that beyond the shallow water areas tides dominate the sediment transport regime, which will be the focus of this study.

### 2.3.2 Regional (Atoll Scale Data)

– **North Male' Atoll and Felidhe' Atoll**

The coral growth patterns in North Male' atoll and Felidhe' atoll studied by Ciarapica and Passeri (1993) were observed to have hard bottoms in areas exposed to high currents, such as the oceanward rim of the atolls and channels, where the combined tidal and wind currents can be in excess of $2 \, \mathrm{m \, s^{-1}}$ (Owen et al., 2011). Detailed studies of the micro atoll of Rasdhoo (e.g., Klostermann et al., 2014; Klostermann and Gischler, 2015; Gischler, 2006; Parker and Gischler, 2011) identified major bed sediment types with regards to foraminifera and facies that correlate well with current patterns. Areas exposed to major currents such as in channels in the outer rim as well as the flanks of the outer atoll were found to be of hard bottom types, and areas with least exposure to strong currents were found to have the highest concentrations



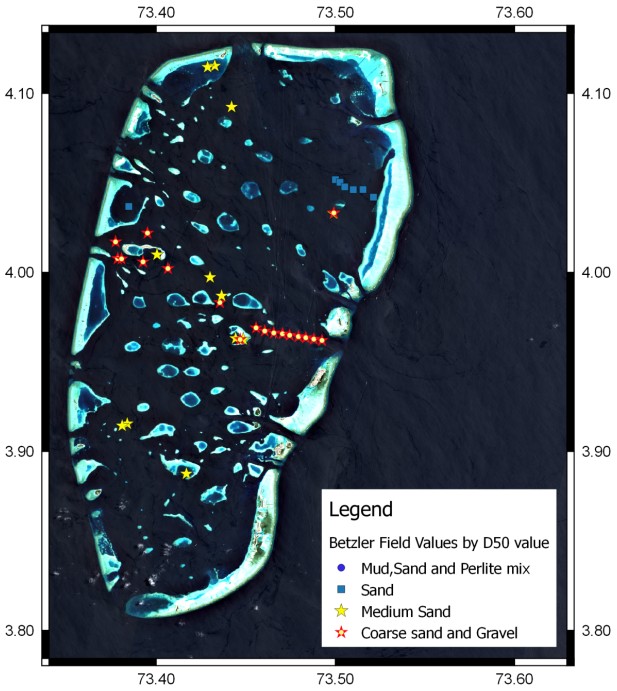

**Figure 2.** $d_{50}$ values for field data obtained by Betzler et al. (2016) laid over Sentinel-2 satellite imagery of South Male' atoll, binned according to grain size classification.

of mud and silt. The association of coral types and growth patterns with varying currents across the atoll indicate a tidally dominated transport pattern across the deep atoll basin (CDE Consulting, 2020).

**– South Male' Atoll**

The most extensive bed sediment study of an atoll within the Maldives archipelago was carried out by Betzler et al. (2015), who collected bed sediment data (Betzler et al., 2016), at different locations of South Male' atoll, Ari atoll and the Maldives inner sea. Importantly, the benthic foraminifera composition at these locations were further classified under five major grain size classes. Figure 2 (a), which illustrates the field data from Betzler et al. (2015), shows that the bed

sediment of the major channels in both the east and west of South Male' atoll is dominated by hard bottom types with a gradual increase of fine particles propagating towards the centre of the atoll. Locations sheltered from major ocean currents, such as the sheltered area in the north-east of South Male' atoll, are dominated by a mixture of mud, pelite (fine fragments of sedimentary rocks) and sand with a small presence of coarse granules. Other areas show a gradual mixture of particles influenced by the velocity patterns arising due to variations in bathymetry. Regions of the atoll basin in the

periphery of high flow regions such as around the drowned lagoons found in the central parts are dominated by medium sand deposited during peak flow and ebb.





## 3 Methods

### 3.1 Tidal Model

In this study we use the Thetis coastal ocean model, a 2D (Angeloudis et al., 2018) and 3D (Kärnä et al., 2018; Pan et al.,
2019) flow solver constructed using the Firedrake finite element solver framework (Rathgeber et al., 2016). Here we use the
2D implementation of Thetis which solves the depth-averaged nonlinear shallow water equations in non-conservative form,
given by

$$\frac{\partial \eta}{\partial t} + \nabla \cdot (H_d \mathbf{u}) = 0, \tag{1}$$

$$\frac{\partial \mathbf{u}}{\partial t} + \mathbf{u} \cdot \nabla \mathbf{u} - \nu \nabla^2 \mathbf{u} + f \mathbf{u}^\perp + g \nabla \eta = -\frac{\tau_b}{\rho H_d}, \tag{2}$$

where $\eta$ is the free surface displacement (m), $H_d$ is the total water depth (m), $\mathbf{u}$ is the depth-averaged velocity vector $(\mathrm{m\,s^{-1}})$
comprising $u$ and $v$ in the $x$- and $y$-directions respectively, and $\nu$ is the kinematic viscosity of the fluid $(\mathrm{m^2\,s^{-1}})$. The term
$f\mathbf{u}^\perp$ accounts for the Coriolis force, where $f = 2\Omega \sin(\zeta)$, with $\Omega$ the angular rotation of the Earth, $\zeta$ the latitude and $\mathbf{u}^\perp$ the
velocity vector rotated $90°$.

The model uses a discontinuous Galerkin based finite element discretization (DG-FEM), specifically the $P_1^{DG}$–$P_1^{DG}$ finite
element pair where piecewise-linear discontinuous function spaces are used to represent both the velocity and the free surface
prognostic fields. For time-stepping, a Crank-Nicolson approach is applied with a constant time step of $\Delta t$. The model treats
wetting and drying according to Kärnä et al. (2011), which introduces a modified bathymetry $\tilde{h} = h + f(H)$ to always ensure
a positive total water depth, with $f(H)$ defined as

$$f(H) = \frac{1}{2}(\sqrt{H^2 + \alpha^2} - H), \tag{3}$$

where $H$ is the water height, and $\alpha$ is a tunable constant. Bed shear stress $\tau_b$ is implemented through the Manning's $n$ formu-
lation

$$\frac{\tau_b}{\rho} = gn^2 \frac{|\mathbf{u}|\mathbf{u}}{H_d^{1/3}}. \tag{4}$$

All simulations were carried out for the period of 00:00 $1^{st}$ January 2018 to 24:00 $5^{th}$ January 2018, which corresponded to
a spring tide in the region. A further 2.5 days of simulation was included at the start to allow for the model dynamics to spin up
from a state of rest. The wetting and drying constant $\alpha$ parameter (Kärnä et al., 2011) was set to 0.5 m and the tidal model was
forced with 11 tidal constituents (M2, S2, N2, K2, O1, P1, Q1, M4, MS4 and MN4) at the open boundaries interpolated from
the TPXO database (Egbert and Erofeeva, 2002). Harmonic analysis of long-term tide gauge data of the region have identified
these constituents as having significant contributions to the combined tidal amplitude in the region (Rasheed et al.).





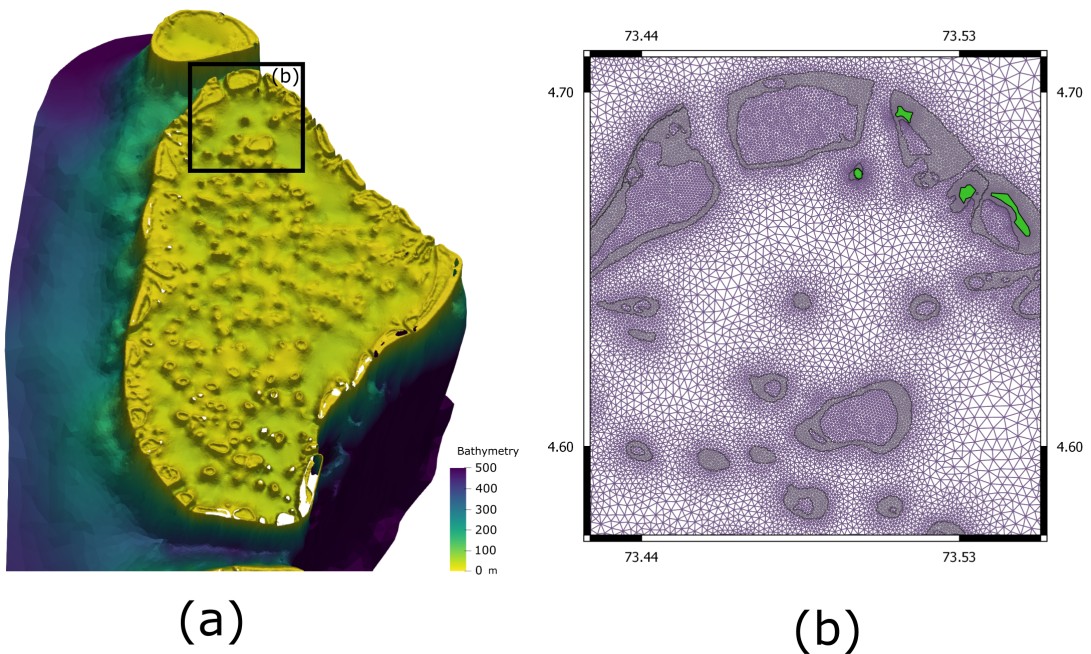

(a)    (b)

**Figure 3.** (a) Bathymetry of North Male' and Gaafaru atoll interpolated onto a simulation mesh. Bathymetry is exaggerated in the vertical for better visualisation, and shows that the individual complex features of the atoll are well captured. (b) Part of the unstructured mesh used for the simulations conducted in this work, focused on northern North Male' atoll. Mesh resolution is increased significantly at lagoons and coastlines.

## 3.2 Model Setup

### 3.2.1 Bathymetry


The complex bathymetry of the Maldives archipelago is not captured by any existing (global) bathymetry datasets; until recently, this has precluded regional (atoll) scale high-resolution modelling. However, recent developments (Rasheed et al.) using satellite data, navigational charts and other sources have produced accurate bathymetry datasets at very high resolution for the first time, facilitating such studies as presented here. Given the scale of the simulations carried out in this study, we use the highest available spatial resolution bathymetry dataset of 0.35 arc seconds ($\approx$10 m). Figure 3 shows the bathymetry of North Male' atoll and Gaafaru atoll interpolated onto a mesh used for the simulations conducted here. The complex features of the atoll, including the narrow channels of the outer rim and the numerous lagoons within the atoll, are well captured.

### 3.2.2 Coastline Data

To setup the simulations used to study the response of coastline modification, next we obtained the coastlines for the corresponding time span. Coastline data for 2018 was chosen as the present day coastline and 1997 was chosen as an 'unmodified'





coastline. Coastline data for 1997 was selected in this context because major reclamation works in the atoll in addition to Male' island, such as the reclamation of the lagoon of Hulhule' to create the artificial island of Hulhumale', began soon after.

Coastline data for the 2018 simulation was obtained from the GADM database (Areas, 2018). However, where this data did not represent the latest changes to the coastlines, Band8A (Narrow infrared) Sentinel-2 satellite images were used to extract
the coastlines, particularly at Hulhumale', Male' and several newly reclaimed islands at the north-west and north-east of the atoll, to provide a better representation of the coastline of North Male' atoll as of March 2018.

For simulations conducted based upon the 1997 coastline data, the coastlines were extracted from Landsat 5 satellite imagery. The single tile LT05_L1GS_145057_19970203_20170102_01_T2 was used, which provides a near cloudless image of the domain area, captured on 3$^{rd}$ March 1997. While other sources provide coastline data across the Maldives archipelago to
varying degrees of accuracy, we found extraction of coastline contours from satellite imagery to be the best way to handle extremely complex and fragmented coastlines such as those in the Maldives.

### 3.2.3 Bed Friction

A uniform Manning drag coefficient was applied across the domain. According to various studies (e.g., Rosman and Hench, 2011), drag parameters across coral reefs are poorly understood and depend on many factors, requiring further study. Given
sufficient observation data, it may be possible to perform a model calibration exercise for the uncertain coefficient (Warder et al., 2020), but this is not conducted here. Instead, the commonly applied value of 0.025 sm$^{-1/3}$ was used. This results in a quadratic drag coefficient ($C_D$) varying from ∼0.0005 in open ocean regions up to ∼0.026 for the shallow reefs and lagoons across the domain, and is consistent with studies reporting the drag coefficient ($C_D$) within reef environments, composed of coral reefs and shallow lagoons (Kraines et al., 1998).

### 205  3.2.4 Unstructured Mesh Generation

The mesh for the model was set up using 'qmesh', a Python package for constructing flexible unstructured meshes for geophysical models (Avdis et al., 2016, 2018) which utilises the 'Gmsh' mesh generator (Geuzaine and Remacle, 2009). The use of unstructured meshes offers significant advantages in representing small spatial features across large geographical extents due to their flexibility over resolution and geometry (Piggott et al., 2008). The meshes that were used for this work were selected
based on a sensitivity study which is described in Section 3.3 and Table 2. For all meshes, the element size at the coastlines was fixed at 50 m, with slightly coarser refinement at reef and lagoon boundaries, and with elements allowed to gradually increase in size to the open boundary. Given the varying distances to this boundary, the element sizes there ranged from 500 m to more than 2000 m. All meshes used for the simulations were generated in the UTM43N coordinate reference system.

Table 1 provides a summary of the meshes and simulation scenarios performed in the study, following the initial sensitivity
study. The triangular element size ranged from 50 m at the island boundaries to up to 2300 m at the open boundary, with refinement to 100 m at the lagoon (reef) boundaries for all three scenarios.





| Simulation | No. of Nodes | No. of Elements | Coastline | Sea Level |
|:---:|:---:|:---:|:---:|:---:|
| 1 | 190,200 | 380,574 | 2018 | MSL |
| 2 | 184,669 | 369,486 | 1997 | MSL |
| 3 | 190,200 | 380,574 | 2018 | MSL+2.0 m |

**Table 1.** Summary of the different meshes and scenarios simulated in the study.

## 3.3   Model Sensitivity

| No. of Nodes | No. of Elements | Resolution at Lagoon (m) | Distance to Boundary (km) |
|:---:|:---:|:---:|:---:|
| \multicolumn{4}{c}{**Sensitivity to Resolution at Lagoon**} | | | |
| 190,200 | 380,574 | 100 | 20 |
| 82,243 | 164,660 | 300 | 20 |
| 74,138 | 148,450 | 500 | 20 |
| 72,356 | 144,886 | 700 | 20 |
| 71,788 | 143,750 | 900 | 20 |
| \multicolumn{4}{c}{**Sensitivity to Distance to Boundary**} | | | |
| 82,243 | 164,660 | 300 | 20 |
| 81,772 | 163,718 | 300 | 15 |
| 80,983 | 162,140 | 300 | 10 |
| 79,618 | 159,410 | 300 | 5 |
| 76,074 | 152,322 | 300 | 2 |

**Table 2.** Summary of the different meshes used for the study. The mesh resolution at the coastlines was set at 50 m and permitted to gradually increase in size up to 15 km towards the open boundaries.

In order to select the most appropriate domain extent (i.e. distance to the open forcing boundary) and mesh resolution pattern in space, a sensitivity study for these choices was conducted through a series of numerical experiments, using the meshes in the configurations summarised in Table 2. The correlation of the simulation data to observations and between simulations was studied through the use of the correlation coefficient $R$ defined as





$$R = \frac{\sum_{n=1}^{N}(O_n - \bar{O})(M_n - \bar{M})}{\sqrt{\sum_{n=1}^{N}(O_n - \bar{O})^2 \sum_{n=1}^{N}(M_n - \bar{M})^2}} \qquad (5)$$

where $N$ is the number of data points, $O_n$ and $M_n$ are the observed and modelled values and $\bar{O}$ and $\bar{M}$ are the means of the observed and modelled values respectively. As described below, initially the correlation coefficient was used to identify

the appropriate resolution of the mesh in order to represent the bathymetry within the domain, after which sensitivity of the simulated tidal amplitude to mesh resolution and distance to the open boundary was investigated.

The correlation coefficient is indicative of the linear least squares fit between the modelled and observed values and has been used widely for sensitivity studies of ocean models (O'Neill et al., 2012). The square of the coefficient, $R^2$, provides a measure of the variation between the observed and modelled values ranging from 0 for no correlation up to 1 for perfect correlation.

### 3.3.1   Mesh Resolution

Since the bathymetry of the domain is highly complex and dominates the flow patterns in the region (Rasheed et al.), assessment of each mesh's ability to faithfully represent the bathymetry is a crucial first step. To select the most appropriate mesh resolution, we use linear interpolation to evaluate the bathymetry on each unstructured meshes at all data points in the high-resolution bathymetry dataset as summarised in Table 2. As highlighted earlier, the resolution at the lagoon varied and the mesh

resolution at the coastline was maintained at 50 m and the mesh element size was allowed to increase up to to a maximum of 15 km in open regions. The correlation between the interpolated bathymetry and the high-resolution bathymetry surface for all data points is given in Figure 4 (a), for varying resolutions at the lagoon. Discounting meshes which did not achieve a correlation of 90%, the meshes which correspond to resolutions of 500 m, 300 m and 100 m at the lagoon boundaries were then used to study the sensitivity of the resulting model outputs to mesh resolution (Figure 4 (b)).

The results of the sensitivity testing of model outputs presented in Figure 4, shows the correlation between the modelled and observed data at the tide gauge located in the harbour of Hulhule' Island. The data shows that for meshes with element sizes at lagoons of 300 m and 500 m the correlation coefficient between the tide gauge data and model results are below a threshold $R^2$ value of 0.9. However, at 100 m resolutions a correlation coefficient of $R^2 \approx 0.9$ is obtained. Based on this result the resolution of the triangular elements making up the multi-scale unstructured mesh was selected to be 100 m at lagoon boundaries.

### 3.3.2   Distance to boundary

Next, to study the model sensitivity to boundary distance, the model was run using the mesh parameters identified in section 3.3.1 but with varying minimum distances to the boundary ranging from 2 km to 20 km as shown in Figure 5 (c). The importance of the distance to boundary has been highlighted in several studies including Jiang et al. (2020), who demonstrated the necessity to include adjacent regions when studying regional scale tidal response to factors such as SLR. Similar issues have been

discussed in the context of altering the tidal dynamics within a boundary forced tidal domain due to the installation of tidal energy devices within a model setup (Adcock et al., 2011), the recommendation being to locate open boundaries beyond the



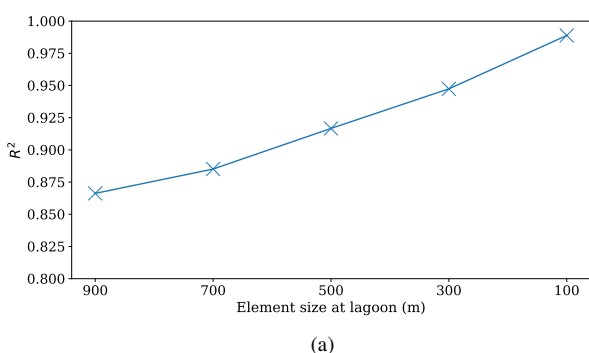
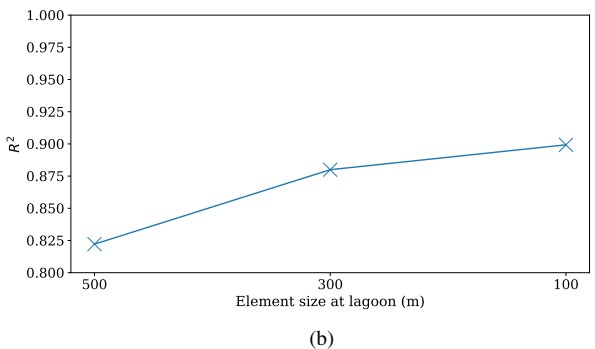

(a)                                               (b)

**Figure 4.** (a) Correlation between bathymetry interpolated onto mesh and the original high-resolution bathymetry dataset for different element sizes at lagoons. (b) Correlation between simulated tidal amplitudes and corresponding tide gauge data for the meshes identified in (a) that achieved $R^2 > 0.9$.

continental shelf in relatively deep water to minimise inconsistencies between the altered dynamics within the domain which are not accounted for in the boundary data.

Initially we compare the correlation between model elevations and tide gauge data, seen in Figure 5 (a), which shows that for all except the simulation with the 5 km minimum boundary distance the correlation coefficient $R^2 > 0.9$. The correlation at 2 km can be attributed to the close proximity to the forcing boundaries. Beyond 5 km there is no significant difference observed in the correlation of tidal amplitudes at the location of the tide gauge with increasing minimum distance to the boundary, with all values corresponding to 10 km, 15 km and 20 km choices showing similarly high correlation values. Since different locations are prone to variability in tidal elevations due to differences in topographic features and other factors (Pugh and Woodworth, 2014), next we compare the tidal elevations at different locations around the atoll.

Figure 5 (b) shows the correlation of tidal amplitudes at three different locations in the domain. The locations, shown in Figure 5 (c), were selected to coincide with channels on the east and west of the atoll which experience the maximum tidal currents, as well as a location in the centre of the atoll. Since there is no observational data available at these locations, we compare the elevations with the results of the simulation carried out using the maximum outer boundary distance of 20 km. For all locations, a very good correlation coefficient is obtained with $R^2 > 0.98$, much higher than the threshold value of $R^2 > 0.9$. However, successive increases in the boundary distance shows a successive increase in correlation with the 10 km and the 15 km boundary results producing results $R^2 > 0.99$ at all locations, providing confidence that 20km is adequate as a boundary distance.

Based on the sensitivity study the mesh resolution at the lagoons (reef) boundaries were selected to be 100 m, and the distance to boundary was set at 20 km. The simulations with the 2018 and SLR configurations were both carried out using 2018 coastlines with a mesh comprising of 190,200 nodes and 380,557 triangular elements, while the mesh used for the 1997 coastline had 184,669 nodes and 369,486 triangular elements.

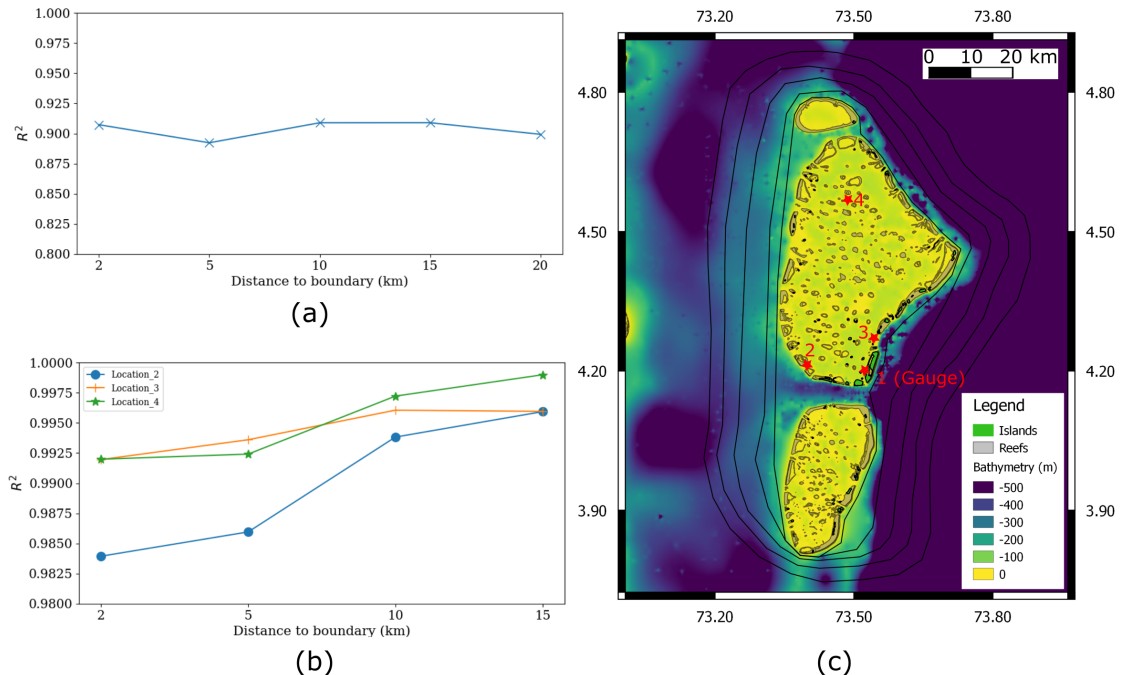

**Figure 5.** (a) Correlation between the simulation amplitudes and tide gauge data (at location 1 shown in (c)) for domains with different minimum distances to the open boundary. (b) Correlation between tidal amplitudes at different locations (labelled 2–4 in (c)) with varying distances to boundary with tidal elevations obtained using the maximum boundary distance at 20 km. (c) Visualisation of the different open boundary locations used for the sensitivity study, the maximum westward boundary follows the 500 m depth contour at the Maldives inner sea where possible. Locations where the elevations are compared are marked in red.

### 3.4 Grain size tidal model proxy

Field data (Betzler et al., 2016), illustrated in Figure 2, and described in Section 2.3.2, was further analysed using the GRADIS-
TAT software (Blott and Pye, 2001) to calculate the $d_{50}$ values, following the procedure adopted by Ward et al. (2015). The $d_{50}$ value, which represents the particle diameter representing the 50% cumulative percentile value, is often used as a representative particle diameter for larger particle groups. Due to the limited number of field points (38 in total), all field points were used.

To develop the grain size tidal proxy, the modelled peak bed shear stress in flood was calculated using equation (4), and is shown in Figure 7 (a). Due to the absence of a large inland mass to obstruct the flow, it is difficult to define the flood and ebb
simply in terms of the reversal of tidal current patterns in coral atolls of the Maldives archipelago; rather we find the rise and fall of water elevations to be more appropriate. While previous studies have used different Manning coefficients for the numerical model and for calibrating bed shear stress for sediment (e.g., Martin-Short et al., 2015), here we find that the same Manning coefficient of 0.025 sm$^{-1/3}$, which provides drag coefficients described in Section 3.2.3, gives results which are comparable to field observations. Next, shear stress values from the model, interpolated at the locations of field data points, were plotted



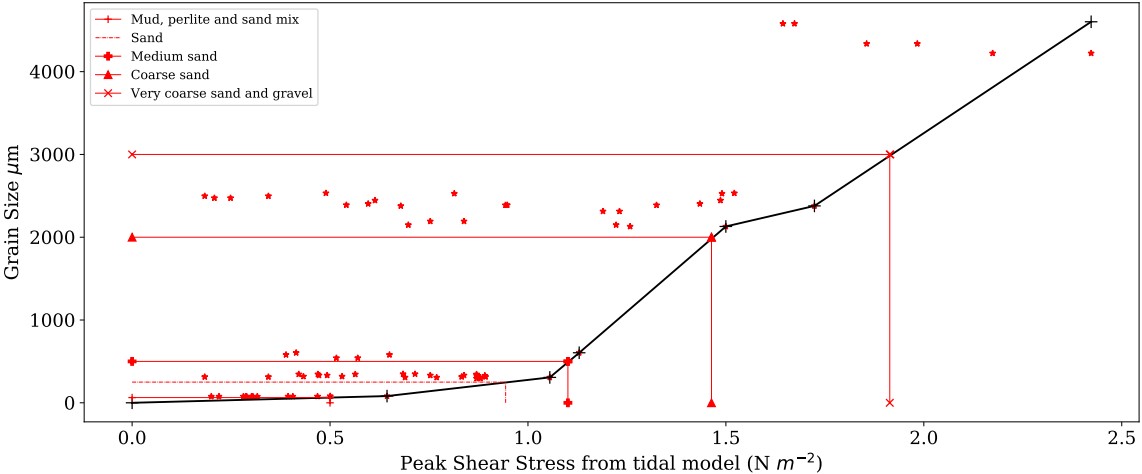

**Figure 6.** Median $d_{50}$ grain size values from Betzler et al. (2016) and associated peak shear stress values from the simulation. A line is fit through the maximum peak shear stress values derived from the simulation, with corresponding peak shear stress values for the grain size classes obtained.

against the calculated $d_{50}$ values from field data; this is shown in Figure 6. Due to the limited number of field observations, a line was fit through the maximum of the bed shear stress values obtained for the simulation. The corresponding peak stress value for the maximum grain size for each sediment class was then obtained as summarised in Table 3, following the grain size classification scheme used in earlier field studies (Betzler et al., 2015). However, given the large values obtained at larger grain sizes in two different clusters, larger grain sizes in the range exceeding 2 mm were further split into two classes.

| Sediment Class | Grain size range ($\mu$m) | Modelled shear stress range (Nm$^2$) |
|---|---|---|
| Mud,Pelite and Sand mix | < 63 | < 0.5 |
| Sand | 63 – 250 | 0.5 – 0.943606 |
| Medium Sand | 250 – 500 | 0.943606 – 1.10092 |
| Coarse Sand | 500 – 2000 | 1.10092 – 1.46362 |
| Very Coarse Sand and Gravel | 2000 – 3000 | 1.46362 – 1.91552 |
| Pebbles and Gravel | > 3000 | > 1.91552 |

**Table 3.** Summary of sediment classes, corresponding grain sizes and modelled bed shear stress values for the developed grain size tidal proxy. The sediment classification is as used by Betzler et al. (2015), the largest grain size class was further split into two (Very Coarse Sand and Gravel and Pebbles and Grave).





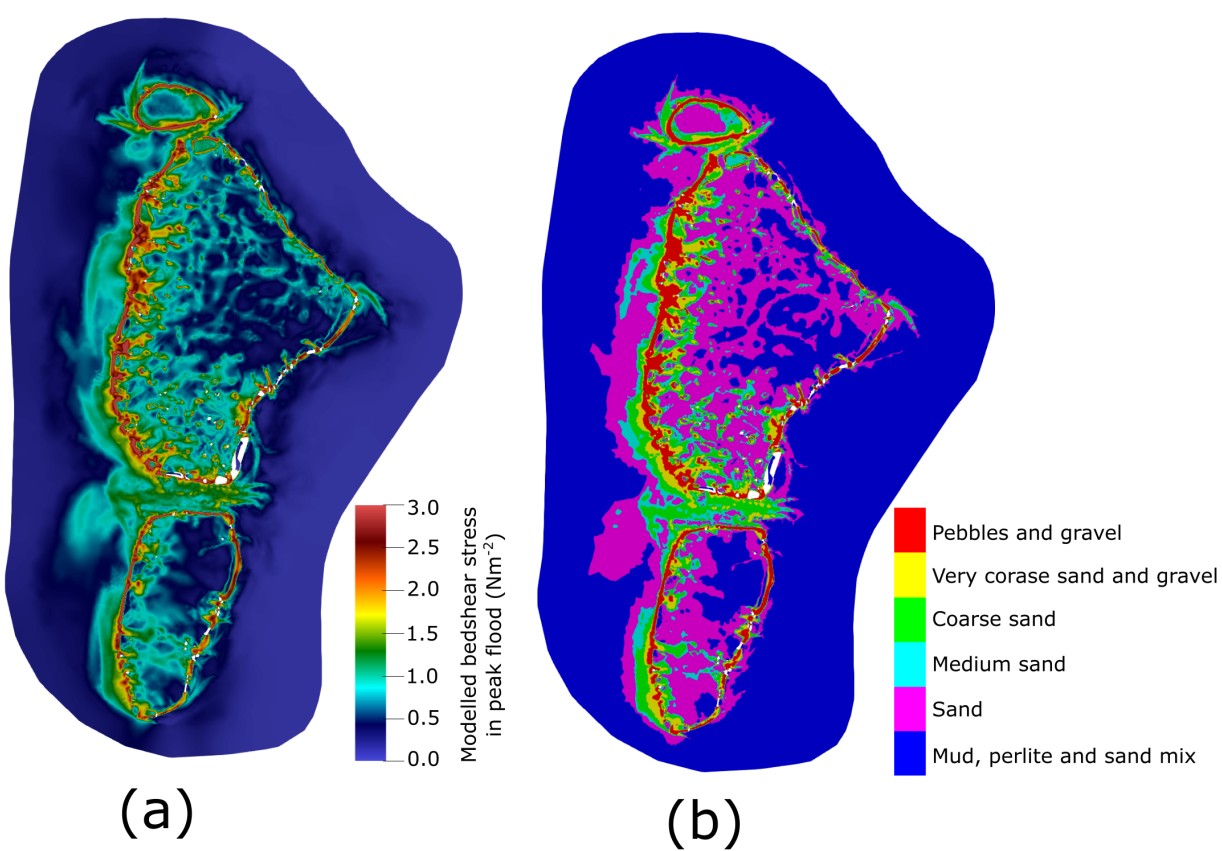

**Figure 7.** (a) Model bed shear stress values at peak flood. (b) Model bed shear stress values binned according to the grain size tidal proxy.

## 4 Results

In this section we apply the grain size tidal proxy developed in Section 3.4 to estimate the current dominant grain size classes across North Male', South Male' and Gaafaru atolls. Further, the changes to the dominant grain sizes across the atoll under different scenarios are derived and discussed.

### 4.1 Grain Size Classification and Validation

The bed shear stress values obtained in the tidal model, and their classification into the dominant sediment classes using the bed shear tidal proxy developed above, are shown in figure 7.

Qualitatively, the model results compare well with the previous studies described in section 2.3. Model results show that the atoll basin is primarily composed of sand. In contrast, regions experiencing high flow velocities, particularly the outer flanks of the atoll rim, the inter atoll channels, openings along the atoll rim (channels), and the flanks of the steep oceanic lagoons





inside the atoll, are comprised of coarser sediment; this is consistent with observations by Kohn (1964); Betzler et al. (2015) and others.

At this resolution, the model also distinguishes sediment particle sizes within individual lagoons, with shallow exposed areas of the lagoons consisting of coarse particles, while the deeper, less exposed areas of individual lagoons are predicted to be comprised of fine particles, in line with observations of Felidhe' atoll (Gischler et al., 2014). Additionally, the presence of

sand on the eastern side of Gaafaru atoll (Gischler et al., 2014) is captured by the model as a region of medium and fine sand pits surrounded by particles of coarser grain size.

In addition, comparison with bed sediment data gathered from environmental impact assessment surveys (CDE Consulting, 2020), which provide a limited bed sediment assessment of the region, indicate that the predicted grain sizes compare well with field data. Comparison of bed sediment type with field data at 56 locations around North Male' atoll shows that the model

predictions matches observations across all locations except for fine sand pits located around the vicinity of islands. This can be attributed to the fact that the model does not incorporate wave-driven sediment patterns which dominate sediment transport in these areas (Kench et al., 2009).

Next, we classified the bottom bed sediment for the simulations carried out using the 1997 coastline scenario, as well as under SLR of 2 m. The same procedure described earlier was adopted and the changes in grain size were compared for each of

the scenarios. Due to the vast majority of the coastal modifications occurring in North Male' atoll (Duvat and Magnan, 2019) we focus on results for this region.

## 4.2    Predicted Bed Sediment Classification Change due to Coastline Modification

Satellite imagery indicates that the landmass of North Male atoll has more than doubled from 10.85 km$^2$ composed of 75 individual islands in 1997, to 26.51 km$^2$ and 88 islands in 2018. Model results, shown in Figure 8, predict that significant changes

in grain sizes across the domain will have occurred as a result of coastline modification over the two decades. In general, the differences in the grain size class increase significantly in the immediate vicinity of areas where significant reclamation has taken place, including the lagoons of reclaimed islands. Significant differences in grain size class can also be seen around channels adjacent to major coastline modifications, arising from increased flow rates. Further, at this scale the model predicts large changes in bed sediment type at the island scale. For example, increased erosion patterns around the island of Kudabandos,

observed since the large scale reclamation of Hulhumale' island (CDE Consulting, 2020), is predicted well in the simulation 8 (a1). The model simulation shows that increased flow rate in the channel between the artificial island of Hulhumale' and the island of Kanifinolhu is the major contributor to the increased erosion, arising from further reclamation of Hulhumale' Island in 2016 completely blocking the flow across the lagoon and forcing the flow through the channel only. This, provides confidence in the model results which predicts similar patterns of change in bed sediment arising from coastline modification. The results

also highlight that the location of the reclamation area is important in influencing the erosion and accretion patterns and the impact of reclamation locations needs to be further studied in major reclamation projects to minimize future damage.

In addition to sediment change driven by artificial reclamation, the results also indicate that tidal sediment in the atoll can be driven by natural changes in the lagoons which are influenced by seasonal wind patterns. Figure 8, shows that the formation of





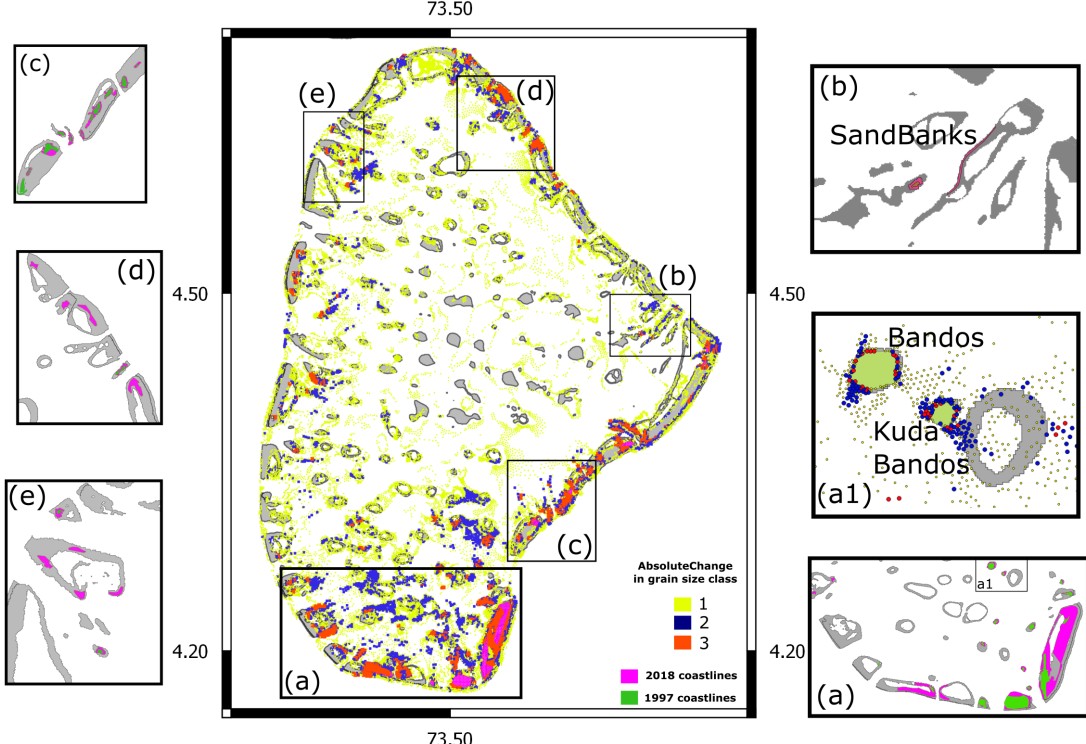

**Figure 8.** Absolute difference in bed sediment grain class predicted to have occurred between 1997 – 2018 due to changes in coastline. Spots coloured 'yellow', 'blue' and 'red' are used to mark locations where an absolute grain size change of 1, 2 or 3 classes has occurred. (a–f) show some of the examples of changes to coastlines which have occurred from 1997 to 2018. (a1) is a zoom of a region shown in (a) showing increased erosion in Kudabandos. (b) Shows the natural formation of sand banks in lagoons which is also predicted to contribute to grain size changes in the atoll basin. The boundaries of reef lagoons (Spalding et al., 2001) are shown in grey for better visualisation. Generally large changes are predicted in the areas where coastlines changes have occurred, but change in grain size is also observed across the wider atoll basin.

sand banks 8 (b) in the *Maabadhi* lagoon of Dhiffushi channel, which are present in the 2018 satellite images but absent in the
satellite image captured in 1997, causes changes in bed sediment in the vicinity of the lagoon. These results predict that natural changes arising from the monsoon wind patterns in the region can influence tidal sediment of the atoll basin in the vicinity of the locations.

### 4.3 Predicted Bed Sediment Classification Change due to SLR of 2 m

While it is clear that global mean sea level is rising (e.g., Church and White, 2006; IPCC et al., 2007; Church et al., 2013), the
extent and rate of sea level rise is the subject of ongoing research. Analysis of long-term tide gauge data (Caldwell et al., 2015)





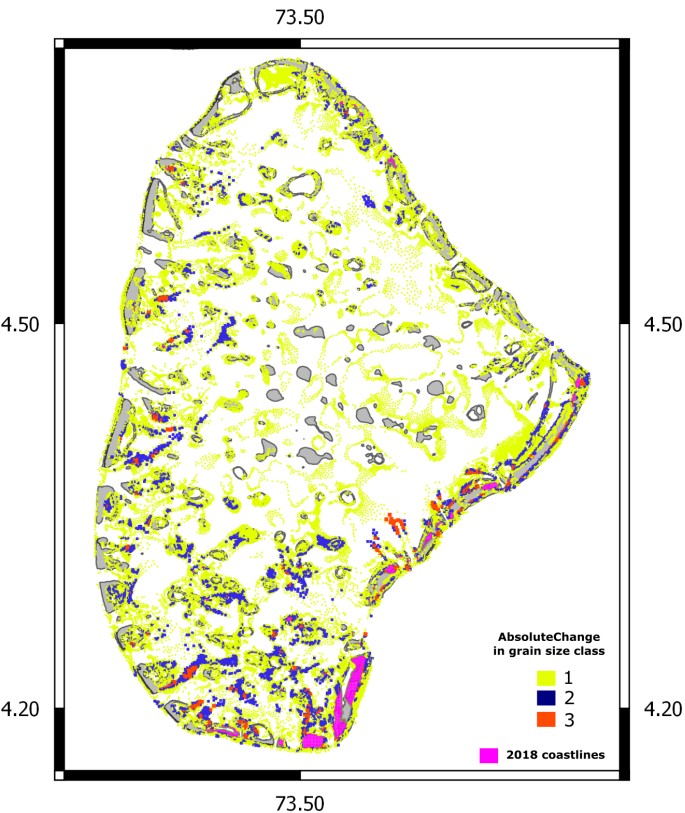

**Figure 9.** Absolute difference in bed sediment grain class arising from a sea level rise of 2 m, with boundaries of lagoons from (Spalding et al., 2001) in grey. Major changes in grain size class is predicted to occur near the channels and the lagoonal faros.

at Hulhule' Island harbour indicates a current local mean sea level rise of ∼4.46 mm per year, with an accelerating trend. Some sources predict a global rise of 0.75 m to 1.9 m by 2100 (Bindoff et al., 2007), while others predict much larger increases in sea level (Vermeer and Rahmstorf, 2009). However, according to studies of glaciological conditions leading to sea level rise, a rise of more than 2 m is unlikely due to kinematic constraints (Pfeffer et al., 2008). Reflecting this, studies have used varying

rates of increase when studying the impact of sea level rise using numerical models. Ward et al. (2012) used a 5 m rise in sea level to study the response of shelf seas to SLR, Pelling et al. (2013) used a SLR of 2 m to study the response of tides in the Bohai Sea and Jiang et al. (2020) used successive rates of SLR up to 2 m to study the response of tides to SLR in a tidal bay. Here we also consider a SLR of 2 m, which represents an upper limit of current predictions of SLR to be reached in a century. This is an important figure as the islands of the Maldives generally have maximum heights of just over 2 m above sea level.

Here we make the simplifying assumption that coastlines remain unchanged during the SLR process. This assumption holds true for many of the current coastlines of North Male' atoll, which are completely or partially protected by artificial barriers.

Similar to the difference in grain size associated with coastline modification, a significant change in bed shear stress is observed with a SLR of 2 m, as seen in Figure 9. No change is predicted by the model at the channel entrances because the bed



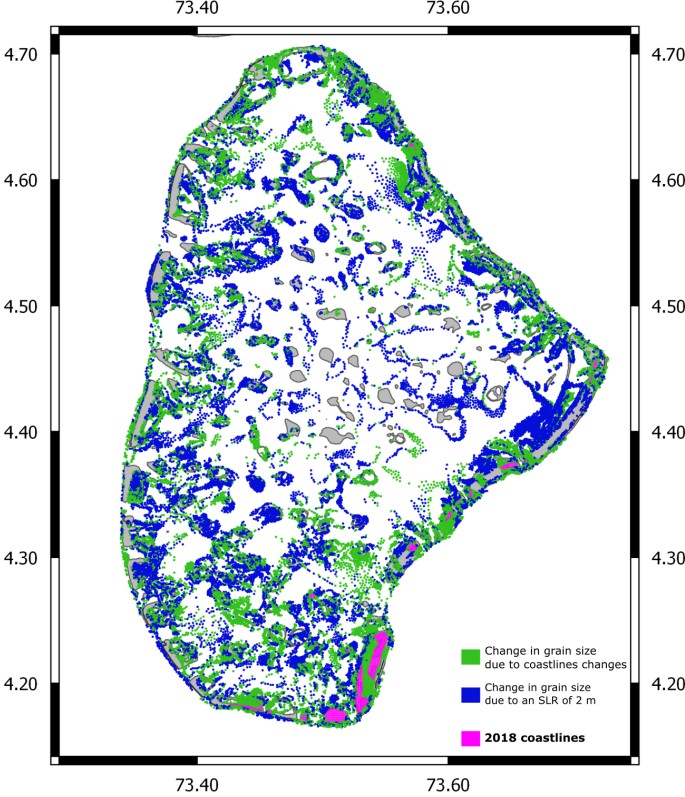

**Figure 10.** Comparison of changes to grain size class with different simulation scenarios, overlaid on boundaries of lagoons from (Spalding et al., 2001) shown in grey. Even though there is generally more grain size class change across the atoll associated with SLR of 2 m, considerable change is predicted with coastline modifications occurring over a far shorter time scale.

sediment at the entrances are already at the coarsest level, but the dominant bed sediment class in the proximity of the major
channels does change, as fine particles are swept away leaving more coarse sediment. The model also predicts wide spread
changes to sediment class at the rims of the lagoons and faros of the inner atoll basin. It is also interesting to note that areas
of significant reclamation such as the southern regions of North Male' atoll are predicted by the model to show an increased
response with SLR. However, this requires further study to be verified and could arise from the change in flow across the region
which has a higher concentration of faros.

## 4.4 Discussion of Results, Limitations of the Study and Recommendations for Improvement

A comparison of the spatial extent of the predicted change in grain size from the two scenarios, shown in Figure 10, indicates
that artificial coastline change over a period of decades produces changes in bed sediment type which are comparable in
magnitude to those due to long-term sea level rise. However, while changes associated with coastline change are more restricted
to the vicinity of the modification itself, the predicted changes associated with SLR are more widespread.





These results agree with studies of other coral atolls around the world (e.g., Webb and Kench, 2010; Duvat and Pillet, 2017; Aslam and Kench, 2017; Duvat and Magnan, 2019), which highlight that the main change driving morphological changes to islands are purely anthropogenic. Further, through the use of a high-resolution hydrodynamic model the results of this study predict that the effects of anthropogenic modifications to island coastlines are not only felt in the vicinity of the lagoons and island coastlines, but are felt across the atoll with potentially far-reaching consequences. This provides further support to

existing studies (Duvat and Magnan, 2019) which show that coastal modification can severely weaken the ability of islands in coral archipelagos to naturally adjust to pressures, increasing their vulnerability to future changes in ocean and climate.

The main limitation of this study is the sparse amounts of available field data. While field data describing grain size for the domain was accessible, the data is relatively scarce. Availability of more field data would enhance the results. Furthermore, tide gauge data was only available for one location within the considered domain and this was located within a sheltered harbour.

Additional tide gauge data would help to further validate the model results and increase confidence in the model set-up and corresponding simulation results.

Additionally, the model results as well as the bed shear stress are highly sensitive to the bathymetry. While this study was made possible by the newly available high-resolution bathymetry dataset (Rasheed et al.), which contains data from a variety of sources collected over decades, it is highly likely that the bathymetry data will contain some errors which impact the final

grain size approximation. To facilitate accurate studies, additional detailed bathymetric data for the region would be beneficial.

Finally, this study focuses on tidally-driven large-scale bottom sediment classification within the larger atoll, with model performance comparable to observational data. To account for the shallow lagoon areas, wind driven sediment can be incorporated into the model, however this requires field data currently not available for model setup and calibration.

## 5    Conclusions

A classified bed sediment map of North Male' atoll, South Male' atoll and Gaafaru atoll has been developed from tidal model simulations. To provide confidence in model results, a sensitivity study was undertaken as well as the model results being compared to observational data. The grain sizes predicted by the model compare well with qualitative and quantitative data reported at the coral atolls of the Maldives archipelago, demonstrating that correctly configured tidal models can be effectively used to determine dominant grain size in coral atolls at the atoll scale. Identification of dominant bed sediment types in coral

atolls can have a wide range of uses ranging from industrial seabed mining to the identification of potential marine flora and fauna habitats.

The response of the estimated bed sediment with coastline changes shows that a significant change in bed grain size distribution occurs at a localised island scale in line with established studies. Using a high-resolution bathymetry dataset coupled with high-resolution hydrodynamic modelling, we have shown here that this change is not limited to the direct vicinity of the

island, but can be seen across the wider atoll basin, consistent with reported observations across the atolls of the Maldives archipelago where significant erosion patterns have been observed in the past few decades without any other major change being reported other than significant island scale modification. While tipping point thresholds for island destabilisation are not

sufficiently well understood (Duvat and Pillet, 2017) to allow predictions for whether these changes are overall detrimental or beneficial in the long term, statistics (DNP, 2019) show that more than 114 of the 198 locally inhabited islands of the Maldives

archipelago (excluding uninhabited and industrial islands) reported severe erosion during the period 2012–2018, endangering long-established island communities and existing socio-economic activities.

Importantly, model predictions show that the change in bed shear grain size associated with a sea level rise of 2 m, which is predicted to occur over timescales ranging from decades to centuries, is comparable to the change in dominant grain size associated with island scale coastline modification which has occurred in the relatively short period of two decades. The results

of this study, derived from detailed numerical modelling, provide support for recent studies which found evidence that artificial coastline modification can be a major factor in increasing the vulnerability of islands (Duvat and Magnan, 2019). With major reclamation work being continuously undertaken in the Maldives archipelago at an industrial scale, and with rates accelerating over the past few years to accommodate major socio-economic activities, the results of this study point to the urgent need for further work to understand the large-scale impacts of coastal modifications.

Further, the results of this study have shown that with recent developments in the availability of high-resolution bathymetry datasets, it is now possible to use hydrodynamic modelling in the Maldives archipelago to study the impact of existing and future coastal modification scenarios at a range of spatial and temporal scales, ranging from open atoll basins to the island scale; these are activities which are generally not currently undertaken in the country (CDE Consulting, 2020). The methods developed in this study can be easily adapted for application to other similar geographic locations, where in general field data

is sparse, and data collection may be hindered for a variety of reasons.

*Code availability.* TEXT

The source code of THETIS coastal ocean model used in this study is available from https://thetisproject.org/ as well as https://github.com/thetisproject/thetis.

*Data availability.* TEXT

**Appendix A**

**A1**

*Author contributions.* TEXT



SR ran the simulations, carried out the analysis of results and initiated the writing of the paper. SCW provided support for model setup and analysis of the results. MDP and YP provided supervision, guidance and insights at every stage of the project.

All authors participated in the writing and editing of the paper.

*Competing interests.* TEXT

The authors declare that they have no conflict of interest.

*Disclaimer.* TEXT

*Acknowledgements.* The authors would like to acknowledge funding from a Research England GCRF award made to Imperial College

London. SR would like to acknowledge PhD funding from the Islamic Development Bank and Imperial College London.



# References

REFERENCE 1

REFERENCE 2



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
