# Peer review of "Response of tidal flow regime and sediment transport in North Male' Atoll, Maldives to coastal modification and sea level rise"

_Ocean Science, 2020_

## Referee Comment (RC1) · Anonymous Referee #1 · 16 Oct 2020

*Overall: This is a very well written and structured paper, and I really enjoyed reading it as it filled a much needed gap in this part of the science. Please may the authors address these two main issues through out the paper, in addition to the points mentioned below. Thank you. - -More connection with oceanographic wind and wave driven processes the explain why some of the changes have occurred. -Revisit assumptions with 2m of SLR.

*Does the paper address relevant scientific questions within the scope of OS? Yes – this paper is well set up and structured. It has clear aims and objectives which are in the scope of the journal and suitable for the wider readership.

*Does the paper present novel concepts, ideas, tools, or data? The paper starts off very descriptive, which is unusual for a journal article to have such a substantial literature review. I think this is OK though as the authors are describing a very specific area that is not known by many oceanographers. Therefore the set up is undertaken well.

Some the reclamation did occur prior to 1997 (e.g. on Male – I think is the 1970s). Also, did beach nourishment for tourist islands occur prior to 1997? Would this affect your model outputs too? In line 240, some additional information on calibration, especially with the tide gauge would be appreciated please.

Fig 6 – it's difficult to see some of the symbols and the difference between then (my eyes are getting older!)

Fig7 – a) It doesn't look like have islands here other than around Greater Male' or it goes over an island. Also applied to Fig 8.

Fig7 – b) Coarseness of scales (e.g. grain size) would be appreciated here please. I don't quite believe all the numbers as on a reality check there is nourishment / lagoons etc given the scale of the information provided. This may not be able to be captured, and if so, a caveat needs to be added.

In Section 4.1 I would welcome some more discussion of the results please. For instance, why is there a lot of pebbles and gravel on the western side of the island? What is driving this? Relating back to the oceanography and wave data would be advantageous. Also, some greater comparisons to the 56 sites. This is glossed over some what, but is an important calibration for the overall dataset.

The start of Section 4.2 could be seen as misleading. Much of this land claim has been on Hulhumale' and then numerous small harbours and airports. When Hulhumale' Phase 2 was constructed, how long was there suspended sediment for? What happens to the eroded sediment? Given the scale of land reclamation in the Maldives, it would be interesting to explain more about the grain size and sediment movement wrt to the

islands which are partly reclaimed (the smaller ones) or have substantial nourishment schemes (i.e tourist islands). For the latter, erosion maybe an issue. These two factors are key development issue and income factor, so understanding this could be a very useful contribution. Furthermore, as the nation is on average 1m above mean sea-level, a 2m rise would make the coastline very, very different. This assumption only considers erosion, but not the fact that some could disappear completely. I know that this depends on the rate of SLR too and contradicts some of Kench's work above, but missing this out puts a big hole in your argument here, and therefore is worthy of greater discussion.

Line 330, the wind driven factors warrants more explanation here please, e,g, duration, permanency, where else seen, influence on grain size, waves (especially long period waves). This would also strength the issues on the discussion / meaning of results as noted two paragraphs above.

Fig 8 – position a1 is unclear on the larger map.

Line 350. Not sure I agree here that coastlines should be static and the work of Kench and others shows this is not the case. Whilst there are numerous islands that are artificial with a sea wall, many are not, as about 20% of the nation is of inhabitable islands (tourist or local). Admittedly, quite a few of these are probably around the capital, but it remains that this is not a valid assumption.

*Are substantial conclusions reached? Yes

*Are the scientific methods and assumptions valid and clearly outlined? Yes, this is really clear throughout. A little more information on the set up and bathymetry would have been welcomed as this comes from a new paper.

*Are the results sufficient to support the interpretations and conclusions? Yes

*Is the description of experiments and calculations sufficiently complete and precise to allow their reproduction by fellow scientists (traceability of results)? Yes – and code

provided (not checked)

*Do the authors give proper credit to related work and clearly indicate their own new/original contribution? Yes

*Does the title clearly reflect the contents of the paper? Yes – but I don't think the SLR was treated well in the paper, so the authors may like to revisit this.

*Does the abstract provide a concise and complete summary? This is probably weaker as there is more about the paper set-up than key results. I didn't learn so much from the abstract itself, so would welcome more results here.

*Is the overall presentation well structured and clear? Yes

*Is the language fluent and precise? Yes

*Are mathematical formulae, symbols, abbreviations, and units correctly defined and used? Yes

*Should any parts of the paper (text, formulae, figures, tables) be clarified, reduced, combined, or eliminated? I am concerned about the SLR being unrealistic as the assumptions made have flaws.

*Are the number and quality of references appropriate? Yes

*Is the amount and quality of supplementary material appropriate? OK.

---

## Referee Comment (RC2) · Anonymous Referee #2 · 16 Nov 2020

The paper reads well and the overall methodology seems ok. I share most the comments adressed by reviewer #1 but would like to add the following:

Wind forcing is either not included in your model or simply not described. From the remaining text I understood the first thing, as e.g. "... wind driven sediment can be incorporated into the model, however this requires field data currently not available for model setup and calibration." However, throughout chapter 2 it is often stated that wind dominates the sediment transport in the study area, e.g. "... studies suggest a wind dominated sediment transport pattern for shallow lagoon areas of the Maldives archipelago". How can you come up with results like "The grain sizes predicted by the

model compare well with qualitative and quantitative data reported at the coral atolls of the Maldives archipelago, demonstrating that correctly configured tidal models can be effectively used to determine dominant grain size in coral atolls at the atoll scale.", if the major driver is not considered? Do you have a tide-only observation available that allows to draw these conclusions? At this point I would expect a more critical and transparent discussion of the issues/drawbacks related to your setup.

You also mention that "The combined tidal- and wind-driven currents can exceed speeds of 2 m s-1." Any idea regarding the magnitude of tide-only induced currents? Maybe this allows to estimate the proportion of sediments transported from tide-only runs.

More specific comments are as follows:

Line 30: Haigh et al. (2019) should be cited as it provides the most comprehensive review on tides and tidal changes... https://agupubs.onlinelibrary.wiley.com/doi/abs/10.1029/2018RG000636

Line 173/177 and elsewhere: add Yr or in review to ref. (Rasheed et al.)

Line 184: check if this sentence is correct. Seems to be wrong... At least I do not get the message!

Line 313: "Next, we classified the bottom bed sediment for the simulations carried out using the 1997 coastline scenario, as well as under SLR of 2 m." Why didi you choose a SLR of 2m? Any justification?

Line 343: "... in global mean sea level"

General comments on the SLR part: The references are rather old and a lot of progress has recently been made regarding SLR projections and individual contributors. You should at least refer to the latest SLR projection from the AR5 and additionally taking recent process understanding (see e.g. Frederikse et al., 2020; https://www.nature.com/articles/s41586-020-2591-3) on ice sheet contributions into

account (see e.g. Edwwards et al., 2019; https://pubmed.ncbi.nlm.nih.gov/30728522/). The entire part on SLR seems a bit antiquated.

Line 348: "...which represents an upper limit of current predictions of SLR to be reached in a century." Says who?

СЗ

---

## Author Comment (AC1) · 17 Dec 2020

**Response of tidal flow regime and sediment transport in North Male' Atoll, Maldives to coastal modification and sea level rise**

**Response to anonymous referee-1**

Shuaib Rasheed [1], Simon C. Warder [1], Yves Plancherel [1,2], and Matthew D. Piggott [1]

[1]Department of Earth Science and Engineering, Imperial College London, UK
[2]Grantham Institute – Climate Change and the Environment, Imperial College London, UK

**Correspondence:** Shuaib Rasheed (s.rasheed18@imperial.ic.ac.uk)

**1 Introduction**

The authors would like to thank the anonymous referee for taking the time to provide a comprehensive review of the submitted manuscript. We are glad that the referee found reading the paper enjoyable, and agrees that the subject is of interest and also of importance, and "addresses a much needed gap in this part of science", particularly in the context of the Maldives, and have not raised any major objection to the presentation, methodology or any other part of the manuscript. We have made the corrections in the manuscript to address each of the comments made by the referee.

Below, we respond to each of the comments from the referee, and provide a summary of the corresponding changes made. Our responses to the comments are presented in blue italics, and any corresponding modifications to the manuscript are presented in framed boxes.

**2 Response to Comments**

The referee raises two main issues:- more connection with oceanographic wind and wave driven processes to explain why some of the changes have occurred and to revisit the assumptions made with respect to 2m of SLR. First we respond to each of these two issues before moving on to address other comments and suggestions.

1. **More connection with oceanographic wind and wave driven processes the explain why some of the changes have occurred.**

*We appreciate that oceanographic wind and wave driven processes drive sediment change in shallow areas, particularly*

*in and around the areas surrounding islands. Since these areas are easy to access, the impact of coastal modification and other factors at these locations have been studied by other authors.*

*However, these areas only cover a relatively small percentage of the wider atoll basin, with sedimentary changes in the deeper atoll basins of the Maldives and elsewhere remaining poorly studied mainly due to inaccessibility. Rather than shallow areas which cover a small area of the larger atoll, and where seasonal winds dominate the sediment transport process, this paper focuses on the deeper atoll basin, which to this day remains poorly studied, and the impact of land reclamation as well as long term SLR on the benthic sediment of the atoll basin which have not been studied before, particularly in the Maldivian context. As field studies are difficult at these depths, we use a validated hydrodynamic model to derive the sediment distribution and predict changes; this approach represents a more viable option which can relatively easily be incorporated into Environmental Impact Assessment Reports for example.*

*Additionally we would like to highlight that this study presents the first atoll scale high resolution numerical modelling study of the Maldives. High resolution modelling of the Maldives archipelago has not been previously possible due to lack of data and has only been made possible due to the availability of a recently developed bathymetry data set, a paper for which is currently at the corrections stage of publication; the inclusion of waves could be considered in a future study. Further, the tidal gauge at Hulhule provides a (limited) means to validate the current tidal model with tidal elevation field data; however, the amount of wave data available in the public domain for any region of the Maldives is currently a significant obstacle in the setup and validation of numerical models studying the the impact of waves.*

2. **Revisit the assumptions with 2m of SLR.**

**Assumption of 2m SLR rate.**

*The referee has raised two issues with the assumption of 2m. Initially with the assumption of 2m SLR rise being unrealistic and the assumptions made regarding the static coastlines of the islands even with a 2m rise given that the islands average height is only around 2m above current sea level.*

*The justification for a 2m SLR assumption has been discussed in detail in several studies, e.g. Jiang et al. (2020), Pelling et al. (2013). Even though a SLR of 5m has been assumed in similar studies Ward et al. (2012), we selected a SLR of 2m as studies suggest 2m as the limit of SLR due to kinematic constraints (Pfeffer et al., 2008). However, there seems to be no agreement as for the time to reach a 2m SLR, and thus we did not delve into this discussion. Additionally, we presented the SLR rate based upon the tide gauge within the domain (Caldwell et al., 2015). As the main objective of the study was to compare potential sediment distribution changes in the atoll basin in the short term due to land reclamation and SLR in the longer term, we feel that the selection of a 2m SLR is a reasonable choice and is discussed at the end of section 3 of the manuscript.*

***Assumption of static coastlines with 2m SLR.***

*Next the referee questions the assumption that the coastline location remain static with SLR. We understand that the justification provided with regards to artificial sea-walls in most of the islands in the considered domain is simplistic. As the referee has highlighted a 2m SLR will obviously modify the coastline, however the future response of the coastlines to SLR is not fully understood, and recent work by Kench et al. (2018) in the Pacific and Cooper et al. (2020) elsewhere suggests that tropical islands such as in the Maldives are more resilient than previously thought, and incorporation of island coastline change would require these datasets to be developed which requires further investigation beyond the scope of this study. Further, the referee points out that reclamation work has been carried out in Male in the 70's. We fully agree with the referee and would like to point out that reclamation work in Male' started not in the 70's but dredging was used to reclaim parts of Male' much earlier than that, with reclamation works taking place in Male' in the early 20th century. However, coastline data for these periods are not easily accessible. We selected the earliest possible satellite imagery to obtain the coastlines, which corresponds to the period just before major reclamation works in Hulhumale' island.*

*Based on this, and as our aim for this study is to compare the bed shear stress response to coastal modification and SLR, particularly in the deeper atoll basin, we feel that the selection of static coastlines is a reasonable, arguably necessary and reproducible choice, with alternative choices introducing many more uncertainties.*

*We have introduced a new paragraph to the manuscript as a subsection summarising the reasons why a SLR of 2 m was selected:*

While it is clear that global mean sea level is rising (e.g., Church and White, 2006; IPCC et al., 2007; Church et al.,

2013), the extent and rate of sea level rise is the subject of significant ongoing research. Analysis of long-term tide gauge data (Caldwell et al., 2015) at Hulhule' Island harbour indicates a current local mean sea level rise of ∼4.46 mm per year, with an accelerating trend, which is in line with recent studies which quote a SLR rate of 3.93 mm per year over the period 1993–2018 for the Indian Ocean and the South Pacific obtained from observations of sea level (Frederikse et al., 2020). Some sources predict a global rise of 0.75 m to 1.9 m by 2100 (Bindoff et al., 2007), while others predict much larger increases in sea level (Vermeer and Rahmstorf, 2009). The Fifth Assessment Report (AR5) of the Intergovernmental Panel on Climate Change (IPCC) proposes that a SLR of 1.0 m is unlikely before 2100 (Church et al., 2013). Additionally, studies of glaciological conditions leading to sea level rise indicate that a rise of more than 2 m is unlikely (Pfeffer et al., 2008). Reflecting this, studies incorporating sea level rise scenarios have used varying rates of increase when studying impacts using numerical models. Ward et al. (2012) used a 5 m rise in sea level to study the response of shelf seas to SLR, Pelling et al. (2013) used a SLR of 2 m to study the response of tides in the Bohai Sea and Jiang et al. (2020) used successive rates of SLR up to 2 m to study the response of tides to SLR in a tidal bay. In line with these studies as well as Bamber et al. (2019), which suggests the use of a global total SLR of 2 m for planning purposes in the 21st century, in this work we also consider a SLR of 2 m. This is an important figure as the islands of the Maldives generally have maximum heights of just over 2 m above sea level. Here we make the simplifying assumption that coastlines remain unchanged during the SLR process. This assumption holds true for many of the current coastlines of North Male' atoll, which are completely or partially protected by artificial barriers. While at other locations this assumption is admittedly hard to justify, we feel this is a reasonable and reproducible choice that avoids the addition of further uncertainties over how the coastlines will respond naturally and anthropologically to SLR.

Next we present the response of each individual comment, highlighting the revisions made to the manuscript which are again presented in framed boxes.

1. Some the reclamation did occur prior to 1997 (e.g. on Male – I think is the 1970s). Also, did beach nourishment for tourist islands occur prior to 1997? Would this affect your model outputs too? In line 240, some additional information on calibration, especially with the tide gauge would be appreciated please.

*We agree with the referee that the land reclamation area presented in section 4.2 is dominated by the land reclamation work at Hulhumale'/Hulhule' island which is the largest reclamation carried out in the country at a single location. However, as presented in Fig 8 we also show that recent land reclamation activity across the atoll creating artificial islands for resort development has a prominent influence at the atoll scale. We have included these coastlines and their influence given the scale of the work and their occurrence across the Maldives, with a focus on the impact of these activities particularly across the atoll basin which is little understood or previously studied. As the coastlines used in the study are the current maximum coastlines of the islands in the domain, and the coastline data from the earliest satellite imagery available, we feel that we have used a reasonable choice. To use intermediate coastlines would require coast-*

*line data which currently does not exist, however as the authors have suggested this would be an interesting future study particularly if the results could be validated with field data.*

*As the referee has suggested the model calibration study requires more additional information we have made the changes below.*

> As described below, we used the correlation coefficient to identify the appropriate resolution for the mesh in order to represent the bathymetry within the domain and also to investigate the sensitivity of the simulated tidal amplitude to mesh resolution and distance to the open (i.e. forcing) boundary.

> Since different locations are prone to variability in tidal elevations due to differences in topographic features and other factors (Pugh and Woodworth, 2014), next we compare the tidal elevations at different locations around the atoll, to understand the sensitivity of tidal elevations at different locations.

95  2. Fig 6 – it's difficult to see some of the symbols and the difference between then (my eyes are getting older!)

*We have uupdated the figure 6 so that the markers are more easy to see and also as the details suggested by the referee are presented in tabulated form in table 3 we have amended the caption of Fig 6, as below :*

> Median $d_{50}$ grain size values from Betzler et al. (2016) and associated peak shear stress values from the simulation. A line is fit through the maximum peak shear stress values derived from the simulation, with corresponding peak shear stress values for the grain size classes obtained (provided in table 3.)

3. Fig7 – a) It doesn't look like have islands here other than around Greater Male' or it goes over an island. Also applied to

100  Fig 8.

*We have amended the caption of Figure 7 as below and directed the reader to refer to Figure 1 for the details of the islands.*

> Figure 7(a) Model bed shear stress values at peak flood. (b) Model bed shear stress values binned according to the grain size tidal proxy. The islands are seen in white and the details of the islands are presented in Figure 1.

4. Fig7 – b) Coarseness of scales (e.g. grain size) would be appreciated here please. I don't quite believe all the numbers as

105  on a reality check there is nourishment / lagoons etc given the scale of the information provided. This may not be able to be captured, and if so, a caveat needs to be added.

*We have added the following to the caption of Figure 7 as a caveat to highlight that the model does not take into account the wind, and this might give rise to discrepancies in shallow areas.*

> As the model does not include the impact of wind driven sediment processes, small differences in grain size across shallow water areas (atoll lagoons) (shown in grey in Figure 1(b) ) might arise.

5. In Section 4.1 I would welcome some more discussion of the results please. For instance, why is there a lot of pebbles and gravel on the western side of the island? What is driving this? Relating back to the oceanography and wave data would be advantageous. Also, some greater comparisons to the 56 sites. This is glossed over some what, but is an important calibration for the overall dataset.

*To highlight the origin of the discrepancy between the bed sediment of the western and eastern lagoons of Gaafaru Island the following explanation was added to the paragraph. As the processes which drive the sediment have never been studied at this scale using hydrodynamic modelling, and as this study does not include wind driven waves, this certainly provides an opportunity for further study.*

> The absence of such sand pits on the western side of the island where the bed sediment is predicted by the model to be dominated by particles of a larger size, correlates well with satellite data. However, as the sediment processes at these relatively shallow depths are influenced by a variety of other factors (Gischler et al., 2014), further work is required to determine the origin of the discrepancy.

*With regards to the 56 sites used for comparison, we have highlighted that only a qualitative comparison is drawn here. A quantitative comparison is difficult because the data from the EIA report presents the benthic sediment data combined with marine benthic composition, making it difficult to compare and contrast. Additionally, we would like to highlight that this data is only used for comparison and not for calibration. Calibration of the model was only carried out using the data from Betzler, which to this date presents the only field measurements of the bed sediment across an atoll of the Maldives, further highlighting the need for more field data as well as the need for modelling approaches such as discussed here which provide an alternative to, but must be used in conjunction with, more expensive field measurements.*

6. The start of Section 4.2 could be seen as misleading. Much of this land claim has been on Hulhumale' and then numerous small harbours and airports. When Hulhumale' Phase 2 was constructed, how long was there suspended sediment for? What happens to the eroded sediment? Given the scale of land reclamation in the Maldives, it would be interesting to explain more about the grain size and sediment movement wrt to the islands which are partly reclaimed (the smaller ones) or have substantial nourishment schemes (i.e tourist islands). For the latter, erosion maybe an issue. These two

factors are key development issue and income factor, so understanding this could be a very useful contribution. Furthermore, as the nation is on average 1m above mean sealevel, a 2m rise would make the coastline very, very different. This assumption only considers erosion, but not the fact that some could disappear completely. I know that this depends on the rate of SLR too and contradicts some of Kench's work above, but missing this out puts a big hole in your argument here, and therefore is worthy of greater discussion.

7. Line 350. Not sure I agree here that coastlines should be static and the work of Kench and others shows this is not the case. Whilst there are numerous islands that are artificial with a sea wall, many are not, as about 20% of the nation is of inhabitable islands (tourist or local). Admittedly, quite a few of these are probably around the capital, but it remains that this is not a valid assumption.

*We have provided our response to the issues raised in points 6 and 7 earlier as these issues were raised to be revisited by the referee. To summarise, given the lack of coastline data over the past century, we believe that the use of coastline from the earliest accessible satellite imagery is a sensible choice, further supported by the fact that the reclamation carried out earlier is negligible in comparison to the reclamation carried out after 1997. Further, as shown in zoom-ins in Figure 8 of the manuscript, the reclamation is not restricted to Hulhumale' but is also extensive across the atoll. Nationwide, as highlighted in the manuscript, studies show that 114 of the 198 inhabited islands have artificially modified coastlines with most islands reclaimed.*

*Further, with respect to the issue of static coastlines, as highlighted earlier we appreciate that the coastlines will be subject to a lot of changes with a 2 m SLR. However, to consider these changes to the coastlines, the topography of the islands as well as the responses of the islands to SLR needs to be studied. While Kench (Kench and Brander, 2006) has shown that the islands of the Maldives are dynamic, a well-known fact, Kench (Kench et al., 2015) also suggests that tropical islands are more resilient to changes caused by SLR. Additionally, in future studies we will consider the impact of wind as well as the potential impact on the island coastlines with SLR.*

8. Line 330, the wind driven factors warrants more explanation here please, e,g, duration, permanency, where else seen, influence on grain size, waves (especially long period waves). This would also strength the issues on the discussion / meaning of results as noted two paragraphs above.

*We have added a new paragraph below to present the reasons why we did not consider wind in this study. However, as highlighted earlier we plan to consider the impact of wind in future studies, where the impact on lagoons will be considered.*

The impact of wind driven sedimentary processes has been discussed in several studies, as summarised in section

2.3.1, particularly with respect to attempts to understand the formation of the Maldives, and the presence of faros in the atolls of the Maldives being attributed to the influence of changes in monsoonal wind patterns (Purdy and Bertram, 1993), (Naseer and Hatcher, 2000). However, studies such as Gischler (2006), which attempted to statistically correlate the presence of geomorphological features in the atolls of the Maldives with various different geological parameters, found that wind speed does not have a significant statistical correlation with the abundance of faros and lagoon reefs within the larger atoll basin. Further, the correlation between both the number of lagoonal faros and marginal faro areas with the wind stress were found to be statistically insignificant. Additionally, studies also show that the island shape influences the morphological change of the islands more than wave exposure (Kench et al., 2009). This provides more support to the case that, while winds dominate sedimentary processes in shallow water areas of the atoll (which constitute an overall small proportion of the area of the wider atoll basin, as illustrated in grey in Figure 1(b) ), wind influence decreases at larger depths with the tides being the major driver of sedimentary transport processes at these depths.

9. Fig 8 – position a1 is unclear on the larger map.

*We have amended Figure 8 to show the position of the extract a1 on the larger map.*

10. Does the abstract provide a concise and complete summary? This is probably weaker as there is more about the paper set-up than key results. I didn't learn so much from the abstract itself, so would welcome more results here.

*As the referee has recommended we have amended the abstract as below to reflect more of the results of the work.*

Changes to coastlines and bathymetry alter tidal dynamics and associated sediment transport process, impacting upon a number of threats facing coastal regions, including flood risk and erosion. Especially vulnerable are coral atolls such as those that make up the Maldives archipelago which has undergone significant land reclamation in recent years and decades, and is also particularly exposed to sea level rise. Here we develop a tidal model of Male' Atoll, Maldives, the first atoll scale and multi atoll scale high resolution numerical model of the atolls of the Maldives, and use it to assess potential changes to sediment grain size distributions under sea level rise and coastline alteration scenarios. The results indicate that the impact of coastline modification over the last two decades at the island scale is not limited to the immediate vicinity of the modified island, but can also significantly impact the sediment grain size distribution across the wider atoll basin. Additionally, the degree of change in sediment distribution which can be associated with sea level rise that is projected to occur over relatively long time periods is predicted to occur over far shorter time periods with coastline changes, highlighting the need to better understand, predict and mitigate the impact of land reclamation and other coastal modifications before conducting such activities.

**3  Conclusions**

170   We have made revisions to the manuscript as highlighted by the referee and we hope that these revisions are satisfactory. Further, we have provided responses to the issues raised by the referee particularly in relation to wind driven processes which are not discussed in the paper, as well as the assumptions on SLR and its impact on the coastlines. We hope that the responses are acceptable to the referee.

**References**

175    Bamber, J. L., Oppenheimer, M., Kopp, R. E., Aspinall, W. P., and Cooke, R. M.: Ice sheet contributions to future sea-level rise from structured expert judgment, Proceedings of the National Academy of Sciences, 116, 11 195–11 200, 2019.

Bindoff, N. L., Willebrand, J., Artale, V., Cazenave, A., Gregory, J. M., Gulev, S., Hanawa, K., Le Quere, C., Levitus, S., Nojiri, Y., et al.: Observations: oceanic climate change and sea level, 2007.

Caldwell, P., Merrifield, M., and Thompson, P.: Sea level measured by tide gauges from global oceans–the Joint Archive for Sea Level
180    holdings (NCEI Accession 0019568), Version 5.5, NOAA National Centers for Environmental Information, Dataset, Centers Environ. Information, Dataset, 10, V5V40S7W, 2015.

Church, J. A. and White, N. J.: A 20th century acceleration in global sea-level rise, Geophysical research letters, 33, 2006.

Church, J. A., Clark, P. U., Cazenave, A., Gregory, J. M., Jevrejeva, S., Levermann, A., Merrifield, M. A., Milne, G. A., Nerem, R. S., Nunn, P. D., et al.: Sea level change, Tech. rep., PM Cambridge University Press, 2013.

185    Cooper, J., Masselink, G., Coco, G., Short, A., Castelle, B., Rogers, K., Anthony, E., Green, A., Kelley, J., Pilkey, O., et al.: Sandy beaches can survive sea-level rise, Nature Climate Change, pp. 1–3, 2020.

Frederikse, T., Landerer, F., Caron, L., Adhikari, S., Parkes, D., Humphrey, V. W., Dangendorf, S., Hogarth, P., Zanna, L., Cheng, L., et al.: The causes of sea-level rise since 1900, Nature, 584, 393–397, 2020.

Gischler, E.: Sedimentation on Rasdhoo and Ari Atolls, Maldives, Indian Ocean, Facies, 52, 341–360, 2006.

190    Gischler, E., Storz, D., and Schmitt, D.: Sizes, shapes, and patterns of coral reefs in the Maldives, Indian Ocean: the influence of wind, storms, and precipitation on a major tropical carbonate platform, Carbonates and Evaporites, 29, 73–87, 2014.

IPCC, C. C. et al.: The physical science basis. Contribution of working group I to the fourth assessment report of the Intergovernmental Panel on Climate Change, Cambridge University Press, Cambridge, United Kingdom and New York, NY, USA, 996, 2007, 2007.

Jiang, L., Gerkema, T., Idier, D., Slangen, A., and Soetaert, K.: Effects of sea-level rise on tides and sediment dynamics in a Dutch tidal bay,
195    Ocean Science, 16, 307–321, 2020.

Kench, P., Parnell, K., and Brander, R.: Monsoonally influenced circulation around coral reef islands and seasonal dynamics of reef island shorelines, Marine Geology, 266, 91–108, 2009.

Kench, P. S. and Brander, R. W.: Response of reef island shorelines to seasonal climate oscillations: South Maalhosmadulu atoll, Maldives, Journal of Geophysical Research: Earth Surface, 111, 2006.

200    Kench, P. S., Thompson, D., Ford, M. R., Ogawa, H., and McLean, R. F.: Coral islands defy sea-level rise over the past century: Records from a central Pacific atoll, Geology, 43, 515–518, 2015.

Kench, P. S., Ford, M. R., and Owen, S. D.: Patterns of island change and persistence offer alternate adaptation pathways for atoll nations, Nature communications, 9, 1–7, 2018.

Naseer, A. and Hatcher, B.: Assessing the integrated growth response of coral reefs to monsoon forcing using morphometric analysis of reefs
205    in Maldives, in: Proceedings of the 9th International Coral Reef Symposium, vol. 1, pp. 75–80, 2000.

Pelling, H., Uehara, K., and Green, J.: The impact of rapid coastline changes and sea level rise on the tides in the Bohai Sea, China, Journal of Geophysical Research: Oceans, 118, 3462–3472, 2013.

Pfeffer, W. T., Harper, J. T., and O'Neel, S.: Kinematic constraints on glacier contributions to 21st-century sea-level rise, Science, 321, 1340–1343, 2008.

210    Purdy, E. G. and Bertram, G. T.: Carbonate concepts from the Maldives, Indian Ocean, 1993.

Vermeer, M. and Rahmstorf, S.: Global sea level linked to global temperature, Proceedings of the national academy of sciences, 106, 21 527–21 532, 2009.

Ward, S. L., Green, J. M., and Pelling, H. E.: Tides, sea-level rise and tidal power extraction on the European shelf, Ocean Dynamics, 62, 1153–1167, 2012.

---

## Author Comment (AC2) · 17 Dec 2020

**Response of tidal flow regime and sediment transport in North Male' Atoll, Maldives to coastal modification and sea level rise**

**Response to anonymous referee-2**

Shuaib Rasheed [1], Simon C. Warder [1], Yves Plancherel [1,2], and Matthew D. Piggott [1]

[1]Department of Earth Science and Engineering, Imperial College London, UK
[2]Grantham Institute – Climate Change and the Environment, Imperial College London, UK

**Correspondence:** Shuaib Rasheed (s.rasheed18@imperial.ic.ac.uk)

**1  Introduction**

The authors would like to thank the anonymous referee for taking the time to provide a comprehensive review of the submitted manuscript. We are glad that the referee found that the paper reads well and the overall methodology is ok.

Below, we respond to each of the comments from the referee, and provide a summary of the corresponding changes made.
Our responses to the comments are presented in blue italics, and any corresponding modifications to the manuscript are presented in framed boxes.

**2  Response to Comments**

1. Wind forcing is either not included in your model or simply not described. From the remaining text I understood the first thing, as e.g. ".. wind driven sediment can be incorporated into the model, however this requires field data currently not available for model setup and calibration." However, throughout chapter 2 it is often stated that wind dominates the sediment transport in the study area, e.g. "... studies suggest a wind dominated sediment transport pattern for shallow lagoon areas of the Maldives archipelago". How can you come up with results like "The grain sizes predicted by the model compare well with qualitative and quantitative data reported at the coral atolls of the Maldives archipelago, demonstrating that correctly configured tidal models can be effectively used to determine dominant grain size in coral atolls at the atoll scale.", if the major driver is not considered? Do you have a tide-only observation available that allows to draw these conclusions? At this point I would expect a more critical and transparent discussion of the issues/drawbacks related to your setup.

*We appreciate that oceanographic wind and wave driven processes drive the sediment change in shallow areas, particularly in and around the surrounding areas of islands. Because these areas are easy to access, the impact of coastal modification and other factors at these locations have been studied by different authors. However, here we focus on the wider atoll basin where the tides dominate the flow. We have highlighted this in several areas of the manuscript, and also included this as a caveat in the areas for improvement and also introduced a new paragraph as below to highlight the issue.*

> *The impact of wind driven sediment processes has been discussed in several studies, as presented in section 2.3.1, particularly with respect to the attempts to understand the formation of the Maldives, and the presence of faros in the atolls of the Maldives has been attributed to the influence of the changes in the monsoonal wind patterns (Purdy and Bertram, 1993),(Naseer and Hatcher, 2000). However, studies such as Gischler (2006), which attempted to statistically correlate the presence of geomorphological features in the atolls of the Maldives with various different geological parameters, found that wind speed does not have a significant statistical correlation with the abundance of faros and lagoon reefs within the larger atoll basin. Further, the correlation between both the number of lagoonal faros and marginal faro areas with the wind stress were found to be statistically insignificant. Additionally, studies also show that the island shape influences the morphological change of the islands more than wave exposure (Kench et al., 2009). This provides more support to the case that, while winds contribute significantly to the sediment processes in shallow water areas of the atoll (atoll lagoons) (which constitute a small area of the wider atoll basin, as illustrated in grey in Figure 1(b) ), the wind influence decreases at larger depths and the tides are the major driver of the sediment transport processes at these depths.*

2. You also mention that "The combined tidal- and wind-driven currents can exceed speeds of 2 m s-1." Any idea regarding the magnitude of tide-only induced currents? Maybe this allows to estimate the proportion of sediments transported from tide-only runs.

*Detailed studies of flow velocities for any part of the Maldives is not available in the public domain, making it a significant obstacle to validate the tidal velocities. The tide gauges in the country only record tidal elevations. We quoted one-off figures from the literature which provide indications of the surface current velocities at different locations in the country, and we find that the tidal model can account for a significant percentage of the tidal velocities quoted. However, without validation data we have not quoted the tidal velocities in the manuscript.*

3. Line 30: Haigh et al. (2019) should be cited as it provides the most comprehensive review on tides and tidal changes... https://agupubs.onlinelibrary.wiley.com/doi/abs/10.1029/2018RG000636

*We thank the referee for pointing to this comprehensive reference; we have now quoted this study.*

4. Line 173/177 and elsewhere: add Yr or in review to ref. (Rasheed et al.)
*We have amended this reference.*

5. Line 184: check if this sentence is correct. Seems to be wrong... At least I do not get the message! *We have rephrased the sentence as below as it appears to be confusing. No global coastline dataset provides an accurate depiction of the coastlines of the Maldives, hence we used satellite imagery to derive the coastline data for all islands in the domain for meshing.*

> Coastline data for two model setups, corresponding to present-day and 1997 coastlines, were extracted from a variety of sources, since widely used global coastline data-sets do not accurately capture all islands within the domain, and no coastline data-sets exist with historical data.

6. Line 313: "Next, we classified the bottom bed sediment for the simulations carried out using the 1997 coastline scenario, as well as under SLR of 2 m." Why did you choose a SLR of 2m? Any justification? *We have rephrased and added the following paragraph to justify the use of 2m SLR. Further, as the paper is concerned with comparing the changes to the sediment of the atoll basin arising due to SLR in the long term and coastal modification in the short term, we believe 2m represents a reasonable choice.*

> While it is clear that global mean sea level is rising (e.g., Church and White, 2006; IPCC et al., 2007; Church et al., 2013), the extent and rate of sea level rise is the subject of significant ongoing research. Analysis of long-term tide gauge data (Caldwell et al., 2015) at Hulhule' Island harbour indicates a current local mean sea level rise of ~4.46 mm per year, with an accelerating trend, which is in line with recent studies which quote a SLR rate of 3.93 mm per year over the period 1993–2018 for the Indian Ocean and the South Pacific obtained from observations of sea level (Frederikse et al., 2020). Some sources predict a global rise of 0.75 m to 1.9 m by 2100 (Bindoff et al., 2007), while others predict much larger increases in sea level (Vermeer and Rahmstorf, 2009). The Fifth Assessment Report (AR5) of the Intergovernmental Panel on Climate Change (IPCC) proposes that a SLR of 1.0 m is unlikely before 2100 (Church et al., 2013). Additionally, studies of glaciological conditions leading to sea level rise indicate that a rise of more than 2 m is unlikely (Pfeffer et al., 2008). Reflecting this, studies incorporating sea level rise scenarios have used varying rates of increase when studying impacts using numerical models. Ward et al. (2012) used a 5 m rise in sea level to study the response of shelf seas to SLR, Pelling et al. (2013) used a SLR of 2 m to study the response of tides in the Bohai Sea and Jiang et al. (2020) used successive rates of SLR up to 2 m to study the response of tides to SLR in a tidal bay. In line with these studies as well as Bamber et al. (2019), which suggests the use of a global total SLR of 2 m for planning purposes in the 21st century, in this work we also consider a SLR of 2 m. This is an important figure as the islands of the Maldives generally have maximum heights of just over 2 m above sea level. Here we make the simplifying assumption that coastlines remain unchanged during the SLR process. This assumption holds true for many of the current coastlines of North Male' atoll, which are completely or partially protected by artificial barriers. While at other locations this assumption is admittedly hard to justify, we feel this is a reasonable and reproducible choice that avoids the addition of further uncertainties over how the coastlines will respond naturally and anthropologically to SLR.

7. Line 343: "... in global mean sea level"

   *Amended as suggested.*

8. General comments on the SLR part: The references are rather old and a lot of progress has recently been made regarding SLR projections and individual contributors. You should at least refer to the latest SLR projection from the AR5 and additionally taking recent process understanding (see e.g. Frederikse et al., 2020; https://www.nature.com/articles/s41586-020-2591-3) on ice sheet contributions into account (see e.g. Edwwards et al., 2019; https://pubmed.ncbi.nlm.nih.gov/30728522/). The entire part on SLR seems a bit antiquated. *We have amended and revisited the SLR section completely as per the referee suggestions and included the suggested references as well.*

9. Line 348: "...which represents an upper limit of current predictions of SLR to be reached in a century." Says who? *We have amended and revisited the SLR section completely as per the referee's suggestions.*

**3   Conclusions**

We have made revisions to the manuscript as highlighted by the referee and we hope that these revisions are satisfactory. Further, we have provided responses to the issues raised by the referee particularly in relation to wind driven processes which are not discussed in the paper, as well as the assumptions on SLR and its impact on the coastlines. We hope that the responses are acceptable to the referee.

**References**

Bamber, J. L., Oppenheimer, M., Kopp, R. E., Aspinall, W. P., and Cooke, R. M.: Ice sheet contributions to future sea-level rise from structured expert judgment, Proceedings of the National Academy of Sciences, 116, 11 195–11 200, 2019.

Bindoff, N. L., Willebrand, J., Artale, V., Cazenave, A., Gregory, J. M., Gulev, S., Hanawa, K., Le Quere, C., Levitus, S., Nojiri, Y., et al.: Observations: oceanic climate change and sea level, 2007.

Caldwell, P., Merrifield, M., and Thompson, P.: Sea level measured by tide gauges from global oceans–the Joint Archive for Sea Level holdings (NCEI Accession 0019568), Version 5.5, NOAA National Centers for Environmental Information, Dataset, Centers Environ. Information, Dataset, 10, V5V40S7W, 2015.

Church, J. A. and White, N. J.: A 20th century acceleration in global sea-level rise, Geophysical research letters, 33, 2006.

Church, J. A., Clark, P. U., Cazenave, A., Gregory, J. M., Jevrejeva, S., Levermann, A., Merrifield, M. A., Milne, G. A., Nerem, R. S., Nunn, P. D., et al.: Sea level change, Tech. rep., PM Cambridge University Press, 2013.

Frederikse, T., Landerer, F., Caron, L., Adhikari, S., Parkes, D., Humphrey, V. W., Dangendorf, S., Hogarth, P., Zanna, L., Cheng, L., et al.: The causes of sea-level rise since 1900, Nature, 584, 393–397, 2020.

Gischler, E.: Sedimentation on Rasdhoo and Ari Atolls, Maldives, Indian Ocean, Facies, 52, 341–360, 2006.

IPCC, C. C. et al.: The physical science basis. Contribution of working group I to the fourth assessment report of the Intergovernmental Panel on Climate Change, Cambridge University Press, Cambridge, United Kingdom and New York, NY, USA, 996, 2007, 2007.

Jiang, L., Gerkema, T., Idier, D., Slangen, A., and Soetaert, K.: Effects of sea-level rise on tides and sediment dynamics in a Dutch tidal bay, Ocean Science, 16, 307–321, 2020.

Kench, P., Parnell, K., and Brander, R.: Monsoonally influenced circulation around coral reef islands and seasonal dynamics of reef island shorelines, Marine Geology, 266, 91–108, 2009.

Naseer, A. and Hatcher, B.: Assessing the integrated growth response of coral reefs to monsoon forcing using morphometric analysis of reefs in Maldives, in: Proceedings of the 9th International Coral Reef Symposium, vol. 1, pp. 75–80, 2000.

Pelling, H., Uehara, K., and Green, J.: The impact of rapid coastline changes and sea level rise on the tides in the Bohai Sea, China, Journal of Geophysical Research: Oceans, 118, 3462–3472, 2013.

Pfeffer, W. T., Harper, J. T., and O'Neel, S.: Kinematic constraints on glacier contributions to 21st-century sea-level rise, Science, 321, 1340–1343, 2008.

Purdy, E. G. and Bertram, G. T.: Carbonate concepts from the Maldives, Indian Ocean, 1993.

Vermeer, M. and Rahmstorf, S.: Global sea level linked to global temperature, Proceedings of the national academy of sciences, 106, 21 527–21 532, 2009.

Ward, S. L., Green, J. M., and Pelling, H. E.: Tides, sea-level rise and tidal power extraction on the European shelf, Ocean Dynamics, 62, 1153–1167, 2012.